# A gender-based review of workplace violence amongst the global health workforce—A scoping review of the literature

**Sioban Nelson**[1], **Basnama Ayaz**[1]*, **Andrea L. Baumann**[1], **Graham Dozois**[2]

**1** Lawrence S. Bloomberg Faculty of Nursing, University of Toronto, Toronto, Ontario, Canada, **2** Princess Margaret Cancer Centre, Toronto, Ontario, Canada

* basnama.ayaz@mail.utoronto.ca

**Data Availability Statement:** The manuscript is a scoping review of workplace violence among healthcare professionals based on gender. All the abstracted data pertinent to the manuscript (226

## Abstract

Workplace violence (WPV) impacts all levels of the health workforce, including the individual provider, organization, and society. While there is a substantial body of literature on various aspects of WPV against the health workforce, gender-based WPV (GB-WPV) has received less attention. Violence in both the workplace and broader society is rooted in gendered socio-economic, cultural, and institutional factors. Developing a robust understanding of GB-WPV is crucial to explore the differing experiences, responses, and outcomes of GB-WPV with respect to gender. We conducted a scoping review and report on the prevalence and risk factors of GB-WPV in healthcare settings globally. The review followed the Preferred Reporting Items for Systematic and Meta-Analyses extension for Scoping Reviews (PRISMA-ScR). We registered the scoping review protocol on the Open Science Framework on January 14, 2022, at https://osf.io/t4pfb/. A systematic search was conducted of empirical literature in five health and social science databases. Of 13667, 226 studies were included in the analysis. Across the studies, more women than men experienced non-physical violence, including verbal abuse, sexual harassment, and bullying. Men experienced more physical violence compared to women. Younger age, less experience, shifting duties, specific clinical settings, lower professional status, organizational hierarchy, and minority status were found to be sensitive to gender, reflecting women's structural disadvantages in the workplace. Given the high prevalence and impact of GB-WPV on women, we provided recommendations to address systemic issues in clinical practice, academia, policy, and research.

## Introduction

To achieve universal health coverage by 2030, the World Health Organization (WHO) and the Global Health Workforce Alliance adopted the global human resources for health (HRH) strategy in 2016. A foundational principle of the strategy is to "uphold the personal, employment, and professional rights of all health workers, including safe and decent working environments and freedom from all kinds of discrimination, coercion, and violence" [1], p.3. However,

studies) is tabulated in Table 1 and Table 2. References are also included for all the sources used in the manuscript. All the references/ sources are publicly available.

**Funding:** The authors received no specific funding for this work.

**Competing interests:** The authors have declared that no competing interests exist.

research demonstrates that workplace violence (WPV) is a significant issue impacting safe work environments for healthcare providers, with far-reaching impacts on individuals, healthcare organizations and society [2].

In 2002, the International Labour Office (ILO), the International Council of Nurses (ICN), the WHO, and Public Services International (PSI) launched a joint program aiming to develop a framework and guidelines for the prevention and elimination of WPV in the healthcare sector [3]. The general definition of violence adopted by the framework (2002) is "incidents where staff are abused, threatened or assaulted in circumstances related to their work, including commuting to and from work, involving an explicit or implicit challenge to their safety, well-being or health" [3], p.3. Over time, this definition has been followed by diverse sources [2, 4], including healthcare which is operationalized in this review. While definitions for various forms of WPV vary widely in different legal jurisdictions and various academic and research studies, we have considered definitions for this review listed in S1 Appendix.

Since the implementation of the above framework by ILO, ICN, WHO, and PSI [2002] to address WPV in the health sector, several studies have been conducted on various aspects of WPV, including the prevalence, risk factors [4, 5], and interventions to de-escalate or eliminate WPV. Studies have been conducted in different clinical settings, geographic locations [6–8] and for different populations in the health workforce. Some studies reported the prevalence of WPV in the last 12 months or six months, while others did not specify the time period. Considering the heterogeneity, we included all the studies that reported gender-segregated data for WPV in the health workforce for this review.

Additionally, some studies reported the prevalence of WPV, which is alarmingly high in certain countries and professional groups. For instance, the prevalence of WPV for nurses was 91% in the USA [9] and 72% in China [10]. Higher incidences of WPV were also reported for physicians in India (41%) [11], and in Australia (58%) [12], and both nurses and physicians in Barbados, where nurses were twice (OR = 2,95% CI 1–5) as likely as physicians to experience verbal abuse [13]. While WPV can affect all healthcare providers, it is particularly problematic for women, who dominate the health workforce in most countries [2]. Some studies report differences in prevalence among male versus female healthcare providers [5, 14]. Our initial impression of the literature is that the issue of gender-based workplace violence (GB-WPV) has received little attention in academic and policy literature, and it is to this aspect that we will now turn.

## Influence of gender on workplace violence

Gender-based workplace violence (GB-WPV) is a worldwide issue rooted in a global culture of discrimination driven by socio-economic, cultural, and institutional factors [3]. While gender-based violence can affect people of all genders, it predominantly affects women, who experience discrimination at higher rates than men and are subjected to various kinds of violence in multiple contexts, most often carried out by men [15, 16]. The United Nations Committee on the Elimination of Discrimination against Women (CEDAW) defined gender-based violence as "violence that is directed against a woman because she is a woman or that affects women disproportionately" [17]. In 2002 case studies were conducted in seven countries (Brazil, Bulgaria, Lebanon, Portugal, South Africa, Thailand and Australia) as part of the ILO, ICN, WHO, and PSI joint program on workplace violence. The case studies revealed that more than 50% of the participants across all healthcare providers in each country, regardless of their profession and gender, experienced physical or psychological violence [18]. In more recent studies, WPV has been reported as particularly harmful to women due to their global preponderance in the health workforce and the impact of gendered power relations that

disproportionally impact women [2, 15]. Therefore, addressing the issue of GB-WPV is critical to meaningfully addressing the issue and supporting the recruitment and retention of women in the health professions [2].

We present a scoping review focused on understanding GB-WPV and related aspects of the global health workforce, including midwives, nurses, and physicians. Our preliminary literature review found that most sources understand gender-based violence as violence against women, including the CEDAW; however, GB-WPV affects everyone regardless of where they identify on the gender spectrum. Since most studies in this global review considered gender as binary (men and women; and a few studies [19–21] included non-binary personnel, which was less than 4% of the sample in those studies, in Table 1, they also reported findings for men

**Table 1. Prevalence of various types/forms of workplace violence by gender in different clinical settings and professional categories across world.**

| S. # | Author/s, year | Purpose | Clinical Setting | Professional Category/ies | Sample (Female; Male) | Country/ies | Prevalence of WPV by Type (Female; Male) |
|---|---|---|---|---|---|---|---|
| 1. | Sun et al., 2022 [5] | Gender differences in the prevalence and risk factors of WPV | District and municipal hospitals | Nurses, physicians and tech | 3426 (73%; 27%) | China | WPV (49%; 61%) Physical Violence (1.1%; 1.8%) Verbal Violence (38%;37%) |
| 2. | Newman et al., 2011 [7] | Influence of gender on workplace violence | Rural and urban settings | Midwives, Nurses, Physicians | 297 (69%; 31%) | Rwanda | Verbal abuse: (68%; 32%); Bullying: (66%; 34%) Sexual harassment: (75%; 25%) Physical attacks: (64%; 36%) |
| 3. | Anand et al., 2016 [11] | Magnitude, consequences, risk factors and reporting patterns for WPV | Tertiary care hospital | Medical residents-various departments | 169 (38.5%; 61.5%) | India | WPV (40%; 41%); Threats (58%; 47%) Physical (4%; 16%); Verbal (17%; 81%) |
| 4. | Forrest et al., 2011 [12] | Prevalence of patient-initiated aggression | All urban and rural settings | General practitioners | 804 (51%; 49%) | Australia | Verbal Abuse (56%; 58%); Physical Abuse (4%;7%), Stalking (3%;4%); Sexual Harassment (10%;2.5%) |
| 5. | Abed et al., 2016 [13] | Prevalence and types of WPV and associated factors | Primary Care clinics | Nurses and physicians | 102 (86%; 14%) | Barbados | Any type of violence (71%; 21%) Verbal abuse: (67%; 21%) |
| 6. | Vyas et al., 2022 [14] | Prevalence of WPV and its risk factors | Tertiary Care Hospital | Nurses, Physicians, and others | 157 Sample not sex-segregated | Uttarakhand North India | Overall violence (65%; 35%) Verbal violence (62%; 38%); Bullying (100%; 0%) Physical violence (60%; 40%) |
| 7. | Giglio et al., 2022 [19] | Prevalence and risk factors for gender-based or sexual harassment | Orthopedics | Physicians | 465 (28%; 72%), NB:1 (0.2%) | Canada | Gender-based harassment (98%; 68) Sexual harassment (83%; 71) |
| 8. | Vargas et al., 2021 [20] | Prevalence of gender policing harassment, heterosexist, and racialized sexual harassment | University medical school | Faculty, fellows, residents, and students | 1288 95.6% Cisgender (52%;48%) LGBTQ + identity (4.4%) | USA | Gender Policing harassment from insiders (59%; 27%) and from patients/families (28%; 7%) Heterosexual harassment (26%; 21%) from insiders and patients/families (22%; 22.7%) Racialized Sexual Harassment from insiders (34.6%; 27%) and patients/families (33;36.6%) |

*(Continued)*

**Table 1.** (Continued)

| S. # | Author/s, year | Purpose | Clinical Setting | Professional Category/ies | Sample (Female; Male) | Country/ies | Prevalence of WPV by Type (Female; Male) |
|---|---|---|---|---|---|---|---|
| 9. | Nukala et al., 2020 [21] | Sexual harassment and predictive factors | Vascular surgery | Medical Trainees | 133 (37%; 61%; others 2%) | USA | Sexual harassment (52%; 23%) |
| 10. | Zhu et al., 2022 [27] | Characteristics of WPV and associated factors | Four county-level primary hospitals | Nurses, physicians and others | 2560 (78%; 22%) | Southeast China | Workplace violence (78%; 22%) |
| 11. | Camm et al., 2021 [28] | Bullying and associated factors | Cardiology | Trainees | 1359 (27%; 73%) | UK | Sexist language (14%; 4%) |
| 12. | Feng et al., 2022 [29] | Prevalence and associated factors | GPs | GPs | 4376 (59%; 41%) | China | Any type (49%; 51%) Physical (37%; 63%); Non-physical (49%; 51%) |
| 13. | Kader et al., 2021 [30] | Factors associated with violence | Registered doctors | Physicians | 157 (14%; 86%) | Bangladesh | Violent incidences (14%; 86%) |
| 14. | La Torre et al., 2022 [31] | Prevalence and determinants of WPV and associated sociodemographic | Multiple centers | Nurses, Physicians and others | 3659 (69%; 31%) | Italy | Physical aggression (9%; 13%) Verbal aggression (47%; 47%) |
| 15. | Orlino et al., 2022 [32] | Factors associated with bullying | Vascular surgery | Medical Trainees | 132 (31%; 69%); | USA | Experience of Bullying (36%; 30%) |
| 16. | Urnberg et al., 2022 [33] | Association between physicians' stress attributed to information systems and workplace aggression. | All sectors, public and private | Physicians | 2786 (67%; 33%) | Finland | Overall aggression (73%; 27%) Physical Aggression (69%; 31%) Non-physical Aggression (77% 23%) |
| 17. | Özdamar Ünal et al., 2022 [34] | Relationship between WPV, job satisfaction and burnout | Various healthcare settings | Physicians and others | 701 (68%; 32%) | Turkey | Workplace violence (70%; 30%) |
| 18. | Syed et al., 2022 [35] | Prevalence of aggressive behavior and its effects | Tertiary care hospital | Nurses & Physicians | 339 (72%; 28%) | Saudi Arabia | Occasionally bullied (69%; 31%); Severely Bullied (67%; 33%) |
| 19. | Atinga, et al., 2021 [36] | The nature, scale and consequences of physical assaults | Psychiatric hospitals | Nurses and non-clinical staff | 501 (57.5%; 42.5%) | Ghana | Physical assaults (59%; 41%) Verbal assaults (75%; 65.7) Sexual harassment (70%; 32%) |
| 20. | Rowe et al., 2022 [37] | Prevalence and sources of mistreatment, and its associations with occupational well-being. | Large academic medical center | Physicians | 1505 (49%; 42%); unknown: 143 (9%) | California, USA | Any forms of mistreatment (31.0%; 15.0%) Sexual harassment (8.8%;1.5%) Verbal abuse (28%; 14%) Physical abuse (6%; 4%) |
| 21. | Vidal-Alves et al., 2021 [38] | Lateral violence and its consequences | Public hospitals | Nursing staff | 950 (78%; 22%) | Southeast of Spain | Mean- personal lateral violence (3.06; 3.41) Mean- social lateral violence (1.92; 1.57) Mean -work-related lateral violence (1.51; 1.28) |
| 22. | Farkas et al., 2022 [39] | Sexual harassment and the impact of a low-cost educational intervention | Internal medicine | Medical residents | Pre: (51%; 48%) 1 no Post: (55%; 39%); 3 no resp. | USA | Gender harassment (88.4%; 49%) Unwanted sexual attention (53.5%; 4.9%) Any harassment (93%; 51%). |

(Continued)

**Table 1.** (Continued)

| S. # | Author/s, year | Purpose | Clinical Setting | Professional Category/ies | Sample (Female; Male) | Country/ies | Prevalence of WPV by Type (Female; Male) |
|---|---|---|---|---|---|---|---|
| 23. | Kibunja et al., 2021 [40] | Prevalence and the consequences of WPV | Emergency department | Nurses | 82 (65%; 35%) | Kenya | Physical violence (71%; 29%) Verbal abuse (65%; 35%) Sexual harassment (82%; 18%) |
| 24. | Sabak et al., 2021 [41] | Frequency and types of WPV | ED | Physicians | 362 (38%;62%) | Turkey | Verbal threats (100%; 97%); Sexual harassment (5%;7%) Physical assaults (50%; 57%); Stalking (16%; 30%) |
| 25. | Lall et al., 2021 [42] | Prevalence, types, and sources of workplace mistreatment and suicidal ideation | Emergency | EM residents | 8470 (35%; 62%), Unknown (3%) | USA | Gender discrimination (65%; 9%) Verbal or emotional abuse (32%; 27%) |
| 26. | Homayuni et al., 2021 [43] | Relationship between role conflict, core self-evaluations and negative effect with bullying | Public and private hospitals | Nursing personnel | 321 (87.5; 12.5%) | Iran | Means scores for bullying (31.4%%; 38.8%) |
| 27. | Holzgang et al., 2021 [44] | Experiences of Negative Workplace Experiences (NWPE) | Surgery | Surgeons | 840 (51%; 48%) 13 individuals did not disclose | European Union | Gender discrimination (49%; 11%) Physical NWPE (13%; 4%) Verbal NWPE (58%; 49%) |
| 28. | Gadjradj et al. 2021 [45] | Estimate the extent of issues of mistreatment. | Neurology | Neurosurgeons and neurosurgical residents | 503 (20%80%) | e-survey for conference participants | Gender discrimination (90.2%; 13.0%) |
| 29. | Subbiah et al., 2022 [46] | Incidence and impact of sexual harassment | Oncology | Oncologists | 271 (56%; 44%) | USA | Sexual harassment-peers and superiors (80%; 56%) Sexual harassment-patients and families (67%; 35%) |
| 30. | Albuainain et al., 2022 [47] | Prevalence of bullying | Surgical environments | Physicians | 788 (35%; 65%) | Saudi Arabia | Negative Attitude Questionnaire-R score (42.7; 42.3) |
| 31. | Nøland et al. 2021 [48] | WPV trends during careers and associated factors | 2-cohorts of medical students | Physicians | 893 (56%; 43%) | Norway | Prevalence of WPV T2 = (14.5%; 27.7%); T3 (11.3%; 25.0%); T4 (9.1%; 14.4%); T5 (7.3%; 10.5%) |
| 32. | Papantoniou, 2022 [49] | Frequency of sexual harassment by gender | Greek NHS | Nurses | 1726 (71%; 29%) | Greece | Sexual harassment (67%; 41%). |
| 33. | Swed et al., 2022 [50] | Prevalence of bullying | Graduate medical education | Medical residents and fellows | 276 (58%; 38%); missing (3.6%) | Syria | Bullying (54%; 30%) |
| 34. | Kowalczuk & Krajewska-Kulak, 2017 [51] | Exposure of patient aggression and potential determinants | General Hospital | Medical, nursing, midwifery, and other personnel | 1624 Medical (56%; 44%) Nurses (98%; 2%) Midwives (99.6%; 0.4%) | Poland | Mean scores of patient aggression 0–5 scale Nurses (26.6; 34.9) Physicians (17.8; 19.7) Midwives (12; 10.9) |
| 35. | Najjar et al., 2022 [52] | Prevalence and forms of gender discrimination and sexual harassment | Hospitalists and general practitioners | Medical students and physicians | 1071 (70%; 29.6%), non-binary (0.4) | Switzerland | Discrimination and sexism (32%; 7%) Sexual harassment (19%; 17%) |

(*Continued*)

**Table 1.** (Continued)

| S. # | Author/s, year | Purpose | Clinical Setting | Professional Category/ies | Sample (Female; Male) | Country/ies | Prevalence of WPV by Type (Female; Male) |
|------|----------------|---------|------------------|---------------------------|------------------------|-------------|------------------------------------------|
| 36. | Notaro et al., 2021 [53] | Prevalence and impact of sexual harassment and assaults | Dermatology | Dermatologists and trainees. | 330 (75%; 24%), 4-unknown | USA | Sexual harassment (94%; 52%) Sexual assaults (35%; 15%) SH among female trainees & attendings (86%; 65%). |
| 37. | Schlick et al., 2021 [54] | Types, sources, and associated factors of gender discrimination and sexual harassment | 301 general surgery programs | Medical residents | 6956 (41%; 59%) | USA | Gender discrimination (80%; 17%) Sexual harassment (43%; 22%) |
| 38. | Shahrour et al., 2022 [55] | The moderating role of social support on WPV and stress | Governmental mental health hospitals | Psychiatric nurses | 195 (42%; 58%) | Jordan | Mean scores for WPV (2.91; 3.27) |
| 39. | Lei et al., 2022 [56] | Prevalence, characteristics, and predictors of WPV | Emergency department | Nurses | 20136 (89%; 11%) | China | Any type of WPV (89%; 11%) Physical (85%; 15%); non-physical (89%; 11%) |
| 40. | Hunter et al., 2022 [57] | Prevalence of violence during clinical placements | Higher Education Institution | Nursing students | 138 (92%; 8%) | Scotland, UK | Ever experienced Verbal violence (70%; 67%). Ever experienced Physical violence (72%; 41%) |
| 41. | Conco et al., 2021 [58] | Prevalence of bullying among and associated factors | Academics Faculty of Health Sciences | All academic staff | 515 (70%; 30%) | South Africa | Bullying: (61%; 48%) |
| 42. | Brooks et al., 2022 [59] | The extent to recall experiences during training | Orthopedics | Black orthopedic residents and fellows | 310 Residents (18%; 82%); Attending (22%; 78%) | USA | Micro assaults (65%; 60%) |
| 43. | Dettmer et al., 2021 [60] | Barriers to career progress for women doctors | Cardiology | Physicians | 567 (49%; 51%) | Germany | Experienced sexual harassment (32%; 7%) |
| 44. | Ferrara et al., 2022 [61] | Prevalence of physical and non-physical violence | Academic | Nursing students | 603 (77%; 23%) | Italy | Psychological violence (39%; 22%) Physical violence (9%; 5%) |
| 45. | Guney wt al., 2022 [62] | Determine violent incidents experiences | General surgery | Surgeons | 94 (11%; 89%) | Turkey | Exposure to violence (90%; 62%) |
| 46. | Pendleton et al., 2021 [63] | Prevalence, frequency, and forms of gender-based discrimination | Three academic institutions | Medical residents from 12 programs | 309 (55%; 45%) | Boston, USA | Gender-based discrimination (100%; 69%) |
| 47. | Snavely et al., 2021 [64] | Rates of non-physical WPV based on gender | Emergency medicine (EM) | Resident trainees | 22 (64%; 36%) | New York, USA | Mean number of incidents/shift (3.0; 0.9) |
| 48. | Hock et al, 2021 [65] | Preparedness to respond to sexual harassment before and after a workshop. | ophthalmology, surgery, medicine and others | Medical residents and faculty | 91 (50%; 48%; No response 2%) | Iowa, USA | Recognition for patient-initiated harassment on a 5-point Likert scale before the workshop (4.0; 3.7) and after workshop participation (4.6; 4.5) |
| 49. | Menhaji et al., 2022 [66] | Prevalence of sexual harassment | OBGYN | Trainees | 366 (86.5%; 13.5%) | USA | Sexual harassment (68.7%; 69.6%) |

*(Continued)*

**Table 1.** (Continued)

| S. # | Author/s, year | Purpose | Clinical Setting | Professional Category/ies | Sample (Female; Male) | Country/ies | Prevalence of WPV by Type (Female; Male) |
|---|---|---|---|---|---|---|---|
| 50. | Ruzycki et al., 2022 [67] | Experiences and perceptions of harassment and discrimination | All practicing physicians in Alberta | All physicians | 1087 (47%; 33%); Others (20%) | Alberta, Canada | Harassment and discrimination among White (76%; 52%) Harassment and discrimination among Black, Indigenous, and People of Color (85%;79%) |
| 51. | Abrams et al., 2011 [68] | Experiences of being stalked by their patients | Urban settings | Physicians (Directory of physicians) | 1190 (35%; 61%; unknown 4%) | Canadian urban area | Stalking (13.5%; 10.9%) |
| 52. | Afkhamzadeh, 2019 [69] | Prevalence of WPV and associated factors | Teaching hospital | Physicians & medical students | 321 (55%; 45%) | Iran | At least one type of violence (51%; 68%) |
| 53. | Alhamad et al., 2021 [70] | Occupational violence and the properties of reported abuse | All kinds of hospitals | Physicians | 969 (35%; 65%) | Jordan | Abuse: (55%; 67%) Verbal abuse: (50%; 60%); Physical abuse: (01%; 08%) |
| 54. | Al-Maskari et al., 2020 [71] | WPW prevalence, characteristics, and factors | Emergency department | Nurses | 103 (74%; 26%) | Oman | Physical: (47% 53%) Non-physical: (24%; 76%) |
| 55. | Al. Shamlan et la., 2017 [72] | Prevalence, consequences & associated characteristics | Teaching Hospital | Nurses | 391 (89%; 11%) | Saudi Arabia | Verbal abuse (28%; 50%) |
| 56. | Al.Surimi et al., 2020 [73] | Workplace bullying and associated factors | 4 hospitals in various regions | Nurses, Physicians, and others | 1075 (86%; 14%) | Saudi Arabia | Bullying (66%; 49%) |
| 57. | Arnold et al., 2020 [74] | Opinions regarding sexual harassment (frequency, type of harassment, and reason) | Pediatrics, internal medicine, and surgery Residents | Physicians | 381 (60%; 40%) | USA | Some sort of harassment during both medical school and residency (55.8%; 35.6%) Sexual harassment (83%; 44%) |
| 58. | Balch Samora et al., 2020 [75] | Experiences of discrimination, bullying, sexual harassment, and harassment (DBSH) | American Academy of Orthopedic Surgeons | Surgeons | 926 (67%; 33%) | USA | Overall DBSH: (81%; 35%) Discrimination (84%; 59%) Sexual harassment (54%; 10%) |
| 59. | Bayram et al., 2017 [76] | Prevalence and factors associated with WPV | Emergency | Medical personnel | 713 (39%; 61%) | Turkey | Workplace violence (40%; 60%) |
| 60. | Belayachi et al., 2010 [77] | Frequency of exposure, characteristics, and impact | Hospital-based emergency | Physicians | 60 (37%; 57%) | Morocco | Overall violence (45%; 55%) |
| 61. | Benzil et al., 2020 [78] | Prevalence and severity of sexual harassment and perpetrators | Neurosurgery | Surgeons | 622 (21%; 78%; others 1%) | USA-professional platforms | Sexual harassment (88%;44) |
| 62. | Bhandari et al., 2021 [79] | Experiences and impact of discrimination & harassment | Internal medicine | Hospitalists | 336 (57%;43%) | USA | Discrimination (99%; 29%) Harassment (72%; 36%, |
| 63. | Boafo et al., 2016 [80] | Verbal abuse and sexual harassment and nurses' response. | Hospitals at all levels | Nurses | 592 (79%; 21%) | Ghana | Verbal Abuse (83;17.0%). |
| 64. | Brown et al., 2019 [81] | Sexual harassment and discrimination | Gynecology | Physicians- members of an international society | 907 (59%; 40%; unknown 1%) | USA and non-USA | Gender discrimination (90%; 72%) Harassment (72.6; 53) Sexual Harassment (84%; 14%) |
| 65. | Camargo & Yousem, 2017 [82] | Prevalence of sexual harassment | Radiology | Radiologists and trainees | 401 (26%; 74%) | USA- and others | Sexual harassment (24.4%; 4.4%) |

(Continued)

**Table 1.** (Continued)

| S. # | Author/s, year | Purpose | Clinical Setting | Professional Category/ies | Sample (Female; Male) | Country/ies | Prevalence of WPV by Type (Female; Male) |
|---|---|---|---|---|---|---|---|
| 66. | Cashmore et al., 2012 [83] | Patterns, severity and outcomes of WPV | Correctional health services | Physicians and nurses, and others | 208 incidents | Australia | Workplace violence incidents (66%; 34%) Verbal Abuse (74%, 26%); Physical Abuse (25%; 55) |
| 67. | Cavalcanti et al., 2018 [84] | Prevalence and risk factors of WPV | PHC | Nurses | 112 (95%; 5%) | Brazil | Workplace violence (72%; 83%) |
| 68. | Ceppa et al., 2020 [85] | Extent of Sexual Harassment | Cardiothoracic surgery | Attending surgeons and trainee surgeons | 790 (23%; 75%; Others 2%). | Globally-professional platforms | Sexual Harassment (81%; 46%) attending surgeons) Sexual harassment (90%; 32% among trainees) |
| 69. | Chang et al., 2020 [86] | Sexual harassment and influencing factors | 4 Universities | Senior nursing students | 310 (87%; 13%) | Taiwan | Sexual Harassment (23.3%; 18.4%) |
| 70. | Chatziionnidis et al., 2018 [87] | Prevalence, sources, impact and psychological support | 20 NICUs | Nurses and Physicians | 398 (87%; 13%) | Greece | Bullying (56%; 36%) |
| 71. | Chen et al.,2021 [88] | Status of gender bias, harassment, misconduct, assault and its effects | Plastic surgery | Plastic surgery trainees | 236 (42%; 54%; Others 4%) | Pittsburgh- USA | Had been presented provocative imagery/ words (42%; 32%). Discomfort from sexually oriented jokes (45%; 33%) |
| 72. | Chen et al., 2018 [89] | Incidence and risk factors of WPV | Tertiary teaching hospital | Nurses | 1983 (92%; 8%) | China | Any type of violence (50%; 38%) Non-physical (50%; 47%); Physical (6%; 9%) |
| 73. | Cheng et al., 2021 [90] | Aggression and its association with employee outcomes | National Health Services | Nurses, Midwives | 147 (28%; 18%) | England, UK | Aggression from Patients (26%; 20%) |
| 74. | Cheung and Yip, 2017 [91] | Prevalence of WPV and associated sociodemographic | Members of the Association of Hong Kong | Nurses | 850 (88%; 12%) | Hong Kong, China | Workplace Violence (44%; 48.6%) |
| 75. | Cho et al., 2020 [92] | Differences in verbal abuse experiences based on personal and work-related characteristics and quality of care | Hospital | Early-career hospital nurses | 1171 (86%; 14%) | USA | **Verbal Abuse from patients/family** 1–3 times per month (58%; 49%) Once a week or more (20%; 32.7%) **From Physicians:** 1–3 times per month (38%; 34%) Once a week or more (5%;9%) |
| 76. | Chrysafi et al., 2017 [93] | Incidences of bullying and sex discrimination | Medical profession | Medical personnel | 1349 (46%; 54%) | Greece | **Threatening behaviours** Surgery (39%; 26%); Medicine (25%; 24%) **Sexual Harassment** Surgery (30%; 7%); Medicine (26%; 7%) |
| 77. | Crebbin et al., 2015 [94] | Prevalence and effects of discrimination, bullying and sexual harassment | Surgery | Medical personnel in surgery | 3516 (19%; 81%) | Australia and New Zealand | Overall prevalence of DBSH (72%; 64%) Bullying (58%; 34%); Sexual Harassment (30%; 2%) |

*(Continued)*

**Table 1.** (Continued)

| S. # | Author/s, year | Purpose | Clinical Setting | Professional Category/ies | Sample (Female; Male) | Country/ies | Prevalence of WPV by Type (Female; Male) |
|---|---|---|---|---|---|---|---|
| 78. | Crutcher et al., 2011 [95] | Frequency, type, source, and basis of intimidation, harassment, or discrimination (IHD) during training | Family medicine | Family medicine residency graduates | 242 (53.2%; 46.4%), Not recorded 0.4%) | Alberta, Canada | IHD by gender (48%; 44%) IHD in the form of work as punishment (20%; 38.6%) IHD in the form of privileges/opportunities being taken away (26.7%; 6.8%) |
| 79. | Dal Pai et al., 2015 [96] | Violence and its association with burnout and psychiatric disorders | Hospital setting | Medical, nursing, and other health personnel | 269 (58%; 42%) | Brazil | Exposed to Violence (71%; 52%) |
| 80. | David et al., 2015 [97] | Rates, character, and context of violence | Members of the American Society of Interventional Pain Physicians | Physicians, NPs and others | 330 (22%; 78%) | USA | Physical (11.2%; 15.8%) Verbal (56.3%; 64.4%) |
| 81. | deVasconcellos et al., 2012 [98] | WPV and associated factors | Public hospitals | Nursing personnel | 1509 (87%; 13%) | Rio de Janeiro, Brazil | Verbal violence (69%; 61%) |
| 82. | Dehghan-chaloshtari and Ghoduosi, 2020 [99] | WPV and effective factors to stop it | Hospital | Nurses | 100 (76%; 24%) | Iran | Physical (82.5%; 17.5%) Verbal violence (78.6%; 21.4%) Bullying and mobbing (70.3%; 29.7%) Sexual abuse (66.7%; 33.3%) |
| 83. | Demeur et al., 2018 [100] | Prevalence of aggression, types, and association with the personality | Flemish (Belgian Federal State) | GPs | 248 (60%; 40%) | Belgium | Raising voice (71%; 29%); Scolding (63%; 37%) Verbal intimidation (63%; 37%) Violating privacy (60.5%; 39.5%); Touching (63%, 37%) Grabbing, slapping & kicking (62%; 38%) Sexual intimidation (70%; 30%) |
| 84. | Difazio et al., 2019 [101] | Bullying, its perpetrators and consequences | Diverse healthcare settings | Nursing personnel, Members of the Russian Nurses Association | 438 (97.5%; 2.5%) | Russian Federation | Bullying (97.5%; 2.5%) |
| 85. | el Ghaziri et al., 2019 [102] | Sex and gender role differences in occupational exposures | Correctional settings | Nurses | 107 (75%; 25%) | USA | Workplace Violence Exposure (97%; 95%) |
| 86. | Elston & Gabe, 2016 [103] | Experience and management of violence in daily work | Primary health care | GPs | 697 (37%; 62%) | South-east England, UK | Physical assault (7%; 13%); Threat of harm (8%; 33%) Verbal abuse (78%; 74%) Afraid of becoming a victim of violence (76%;60%) |
| 87. | Falavigna et al., 2021 [104] | Perception of the gender discrimination | Surgery | Spine surgeons | 223 (12%; 88%) | Latin America | Gender discrimination- (66.67%; 1.02%) Sexual harassment- (44.44%; 7.65%) |
| 88. | Ferri et al., 2016 [105] | Frequency, characteristics of WPV and associated factors | General Hospital | Nursing and medical personnel | 419 (67%;33%) | Italy | WPV (assaulted): 45% (72%; 28%) |

(Continued)

**Table 1.** (Continued)

| S. # | Author/s, year | Purpose | Clinical Setting | Professional Category/ies | Sample (Female; Male) | Country/ies | Prevalence of WPV by Type (Female; Male) |
|---|---|---|---|---|---|---|---|
| 89. | Ferri et al., 2020 [106] | WPV and associated factors | Emergency Triage area | Nurses | 27 (44%; 56%) | Italy | Verbal (83%; 100%) Both verbal and physical (17%; 0%) |
| 90. | Fitzgerald et al., 2019 [107] | Prevalence of harassment and discrimination | Academic teaching hospitals | Surgical Residents | 76 (49%; 51%) | USA | At least one form of abuse and harassment (48%; 52.5%). Discrimination in relation to gender (92%; 8%). |
| 91. | Fnais et al., 2013 [108] | Prevalence of harassment and discrimination | Residency programs in teaching hospitals | Trainee residents | 213 (42%; 58%) | Saudi Arabia | Verbal harassment during training (76%; 51%) Gender discrimination (69%; 57%) Sexual harassment (28%; 13%) |
| 92. | Freedman-Weiss et al., 2020 [109] | Prevalence of sexual harassment, characteristics of and barriers to report | Surgical training programs | Trinee residents | 270 (44%; 53%) others; 3% | USA | Sexual Harassment 49% (70.8%; 30.8%) |
| 93. | Fujita et al., 2012 [110] | WPV and the attributes of healthcare staff | Teaching hospitals | Nursing and medical personnel | 8711 (72%; 22%), Not reported; 6%) | Japan | Experience of at least one Kind (40%; 27%) Physical aggression (18%; 10%) Verbal abuse (32%; 24%); Sexual harassment (12%; 4%) |
| 94. | Fute et al., 2015 [111] | Prevalence and associated factors of WPV | Public health facilities | Nurses | 642 (63%; 37%) | Ethiopia | Workplace violence (36%; 20%) |
| 95. | Harthi et al., 2020 [112] | Prevalence of WPV and associated factors. | Public hospitals ED | HCWs in ED, including nursing and medical personnel | 324 (66%; 34%) | Saudi Arabia | Workplace violence (42.8%; 57.8%) Physical violence (11%; 20%); Verbal abuse (35%; 47%) |
| 96. | Hills, 2017 [113] | Differences of exposure to aggression and the risk and protective factors | Clinical medical practitioners | Medical personnel | 9449 (42%; 57%; Missing: 1%) | Australia | Aggression from external sources (69.6%; 66%) Aggression from internal sources (28.9%; 25.9%) |
| 97. | Hills et al., 2012 [114] | Prevalence of aggression from patients and others | Clinical medical practitioners | Medical personnel | 9438 (43%; 57%) | Australia | Verbal or written aggression (72.6%; 69%) Physical aggression (33.8%; 31.2%) |
| 98. | Honarvar et al., 2019 [115] | Prevalence, predictors, and sources of violence | University-affiliated public hospitals | Nursing personnel | 405 (81%; 19%) | Iran | Verbal Abuse (84%; 83%); Verbal threat (25.6%; 46.4%) Physical violence (16.8%; 41.6%) Sexual Harassment (9.8%; 15.6%) |
| 99. | Hsiao et al., 2021 [116] | Sexual harassment across the academic medicine hierarchy | University of Florida College of Medicine | Medical personnel | 509 (54%; 46% | USA | Sexual harassment (46.2%;19.4%) |
| 100. | Hu et al., 2019 [117] | Mistreatment and its association with burnout and suicidal thoughts | General surgery residency programs. | Medical residents | 7409 (39.6%; 59.9%; No data 0.5% | USA | Gender discrimination (65.1%;10.0%). Verbal or emotional Abuse (33.0%; 28.3%) Sexual harassment (19.9%; 3.9%) |

*(Continued)*

**Table 1.** (Continued)

| S. # | Author/s, year | Purpose | Clinical Setting | Professional Category/ies | Sample (Female; Male) | Country/ies | Prevalence of WPV by Type (Female; Male) |
|---|---|---|---|---|---|---|---|
| 101. | Jaijee et al., 2021 [118] | Frequency and types of sexism | Cardiology | Consultant cardiologists | 174 (24%;76%) | The UK | Discrimination of any type (61.9%; 19.7%). Sexual harassment (35.7%; 6.1%) Gender-based discrimination (52.2%; 2.3%) |
| 102. | Jain et al., 2019 [119] | Gender differences in the career and personal profiles | Ophthalmology | Ophthalmologists | 282 (32%;68%) | Australian | Bullying (43%; 33%); Discrimination (31%; 8%) Sexual harassment (23%; 0.5%) Had been excluded-work social events (19%; 1.6%) Humiliating comments (22%; 3%) |
| 103. | Jaradat et al., 2016 [120] | Prevalence of workplace aggression and its consequences | Hospitals and primary care clinics | Nurses | 343 (62%; 38%) | Palestine | Workplace Aggression (26%; 28%) Physical aggression (5%; 5%) Verbal Aggression 24%; 25%); Bullying (5%; 12%) |
| 104. | Kemper & Schwartz, 2020 [121] | Prevalence of WPV and related burnout | Pediatrics | Pediatric residents—Resident Burnout and Resilience Study Consortium | 1956 (70%; 30) | USA | Any type of Mistreatment 33% (36%; 25%) Bullying19% (20%; 16%) Discrimination 18% (21%; 11%) Sexual Harassment 5.4% (6%; 4%) |
| 105. | Kisiel et al., 2020 [122] | Changes in the prevalence and context of self-reported Gender Discrimination and Sexual Harassment between 2002 and 2013. | Uppsala University | Medical students | **2002**–343 (55%: 45%) **2013**–720 (62%; 38%) | Sweden | **2013** (pre-clinical group) Discrimination (22%; 15%); Favoritism (23%; 18%) Intrusive, unwelcome acts (21%; 15%) **2013** (clinical group) Discrimination (41% and 25%); Favoritism (53%; 33%) Intrusive, unwelcome acts (26%; 20%) |
| 106. | Lafta & Falah, 2019 [123] | WPV and its influence on work and life | Hospitals and primary healthcare centres | Medical, nursing personnel and others. | 700 (51%; 49%) | Iraq | Physical violence (24%; 76%) Verbal (53%; 47%) |
| 107. | Li et al., 2021 [124] | Effects of resilience as a mediator in violence toward nurses' intention to leave | Emergency department | Emergency room nurses | 132 (91%; 9%) | Taiwan | Mental violence (54%; 50%) Physical violence (12.5%; 8.3%) |
| 108. | Li et al., 2020 [125] | Gender differences in WPV and its outcomes | General Hospitals Survey | Nursing Workforce | 396 (73%; 27%) | China | Verbal abuse from patients and/families (34%;30%) Verbal abuse from staff (59%; 47%) Physical abuse from patients/families (64%; 58%) Physical abuse from staff (81%; 70%) |

(Continued)

**Table 1.** (Continued)

| S. # | Author/s, year | Purpose | Clinical Setting | Professional Category/ies | Sample (Female; Male) | Country/ies | Prevalence of WPV by Type (Female; Male) |
|------|----------------|---------|------------------|---------------------------|----------------------|-------------|------------------------------------------|
| 109. | Li et al., 2010 [126] | Prevalence of abuse and harassment | Emergency Medicine | Residents | 196 (53%; 47%) | USA | Sexual harassment 23% (37%; 8%). |
| 110. | Lu et al., 2020 [127] | Gender-based discrimination and sexual harassment | Academic Emergency Medicine faculty | Emergency Medicine faculty | 144 (39%; 61%) | England, UK | Discrimination based on gender (62.7%; 12.5%) Unwanted sexual harassment behaviors (52.9%; 26.2% |
| 111. | Lucas-Guerrero et al., 2020 [128] | Professional burnout and contributing factors | General Surgery | Surgical residents | 452 (66%; 34%) | Spain | Physical abuse (5.4%; 10.5%) Sexual harassment (21.4%; 6.5%) Discrimination (90.4%, 9.6%). |
| 112. | Margavi et al., 2020 [129] | Frequency, type, and severity of violence and its consequences during CPR | Emergency wards of teaching hospitals | Nurses | 140 (61%; 39%) | Iran | Physical violence (43%; 76%) Psychological violence (90%; 83%) Sexual harassment (8%; 0%) Bullying/mobbing (34%; 46%) |
| 113. | Martins et al., 2021 [130] | Harassment and its effects on mental health | Surgery | Surgical trainees | 147 (67%; 33%) | Pakistan | Workplace harassment 54.4% (57%; 49%) |
| 114. | McKinley et al., 2019 [131] | Perceived sources, frequency, forms, and effects of Gender-based discrimination | Residency training programs | Medical residents | 371 (46%; 53%); Others 2 | Massachusetts, USA | Gender-based discrimination (93%; 24%). Sexual harassment during training (34%; 5) |
| 115. | Meyer et al., 2021 [132] | Prevalence of discrimination, bullying and Sexual harassment and the scope of action and resolution rates | Ophthalmology | Trainees and Ophthalmologist | **In 2015**–582 (29%; 71%) **In 2018**–560 (29%; 71%) | Australia and New Zealand | Sexual harassment (32%; 4%). Discrimination (43%;12%) Bullying (51%; 31%) |
| 116. | Mirza et al., 2012 [133] | Magnitude, types and the possible etiology of WPV | Emergency department | Physicians in training | 675 (47%; 53%) | Pakistan | Verbal abuse (61; 63%) Physical abuse (8%; 15%) |
| 117. | Moman et al., 2020 [134] | Prevalence and characteristics of WPV | Pain management clinicians | Medical and nursing personnel | 58 (41%; 59%) | Conference participants, USA | Experienced assault (59%; 75%) |
| 118. | Moutier et al., 2016 [135] | Climate concerns among health sciences faculty after implementing an intervention | University of San Diego, health sciences | Faculty members | 478-in 2012 (44%; 29%) 729-in 2015 (54%; 38%) | California, USA | **In 2012** Derogatory comments (34%; 24%) Anger outburst (27%; 19%) Hostile email/ communication (25%; 22%) Intimidating/ bullying behavior (26%; 19%) **In 2015** Derogatory comments (19%; 12%) Anger outburst (22%; 14%) Hostile email/ communication (19%; 13%) Intimidating/ bullying behavior (26%; 13%) |

*(Continued)*

**Table 1.** (Continued)

| S. # | Author/s, year | Purpose | Clinical Setting | Professional Category/ies | Sample (Female; Male) | Country/ies | Prevalence of WPV by Type (Female; Male) |
|---|---|---|---|---|---|---|---|
| 119. | Moylan et al., 2014 [136] | Gender differences in perceptions of and responses to physical assault | Acute care psychiatric facilities | Nurses | 110 (85%; 15%) | New York, USA | Physical assault 73% (80%; 20%) |
| 120. | Nieto-Gutierrez et al., 2018 [137] | Prevalence of WPV and association with the medical specialty | Medical Residency programs | Medical residents | 1054 (38%; 62%) | Peru, South America | Workplace Violence (75%; 72%) |
| 121. | Oguz et al., 2020 [138] | Healthcare workers' states of exposure to violence | Pediatric clinics | Medical and nursing personnel and others | 182 (78.5; 21.5%) | Turkey | Violence (72%; 27%) |
| 122. | Park & Choi, 2020 [139] | Factors influencing being either victims or perpetrators of verbal violence | General Hospital | Nurses | 205 (89%; 11%) | South Korea | Mean experience of Verbal violence (23, 29) Mean Doing Verbal violence (19, 24) |
| 123. | Picakciefe et al., 2017 [140] | Mobbing and its relationship with sociodemographics and work conditions | Primary health care | Health personnel | 119 (83%;17%) | Turkey | Mobbing 31% (89%; 11%) |
| 124. | Pinar et al., 2017 [141] | WPV, the type and structure of the violent incidents | All level healthcare institutions | Health personnel | 12,944 (60%; 40%) | Turkey | Workplace violence in 12 months (48%; 39.5%) Violence during the career (54.3%; 49.4%) |
| 125. | Pol et al., 2019 [142] | Patients' aggressive and violent behaviours since the introduction of the National Emergency Access Target (NEAT) | Intensive Care Unit, | ICU Nurse clinicians | 47 patient records (18 {823 records} pre and 29 {914 records} post-NEAT) | Australia | Verbal violence (20%; 66.7%) Physical violence (45.7%; 25%) |
| 126. | Prajapati et al., 2013 [143] | Security perception and situation of the health workforce | All kinds of health facilities | Medical, nursing, midwifery, and other personnel | 747 | Nepal | Gender-based harassment (62.5%; 37.5%) Sexual Harassment (56.5%; 43.5%) |
| 127. | Rosta & Aasland, 2018 [144] | Prevalence of perceived bullying over time at work in 1993, 2004 and 2014–2015. | Hospital | Medical personnel | **1993** = 2439 (28%; 72%) **2004** = 730 (31.5%; 68.5) **2014–15** = 1080 (43%; 67%) | Norway | Perceived Bullying in: 1993 = (8.3%; 4.7%) 2004 = (4.8%; 8.4%) 2014–15 = (9.2%;5.4%) |
| 128. | Rouse et al., 2016 [145] | Types and frequency of workplace bullying | Academic settings | Family physicians | 1065 (43%; 57%) | USA | Ever displayed bullying behaviors (7.7%; 11.2%) Ever been bullied (34%; 24.7) |
| 129. | Sachdeva et al., 2019 [146] | Incidence and characteristics of WPV and its impact | Emergency Department | Medical and nursing personnel | 335 (34%; 66%) | India | Verbal abuse (37%; 63%); Physical abuse (32%; 68%) Confrontation (12%;88%) |
| 130. | Sakellaropoulos et al., 2011 [147] | Prevalence of workplace aggression and its impact | Anesthesia | Certified registered nurse anesthetists | 205 (62%; 37%) | USA | Verbal aggression (89%; 83%) |
| 131. | Scruggs et al., 2020 [148] | Frequency and severity of sexual harassment | Ophthalmology | Trainees | 112 (47%; 53%) | USA | Sexual harassment from patients (86.8%; 44.1%). Physical harassment (24.5%; 8.5%) |

(Continued)

Table 1. (Continued)

| S. # | Author/s, year | Purpose | Clinical Setting | Professional Category/ies | Sample (Female; Male) | Country/ies | Prevalence of WPV by Type (Female; Male) |
|------|----------------|---------|------------------|---------------------------|----------------------|-------------|------------------------------------------|
| 132. | Sharma et al., 2021 [149] | Global prevalence of a hostile work environment (HWE) and its impact | Cardiology | Cardiologists | 5931 23%; (77%) | Globally | Gender discrimination/ sexual harassment (57%; 22%) Hostile work environment (68%; 37%) Sexual harassment (12%; 1%). |
| 133. | Siller et al., 2017 [150] | Extent of mistreatment by various groups, gender differences and reporting sexual harassment | A medical university | Students | 88 (51%; 49%) | Austria | Harassment and sexual mistreatment (68.9%; 32.6%). Hitting, kicking, or shoving (24.4%; 48.8%). Humiliation (77.8%; 53.5%). |
| 134. | Simoes et al., 2020 [151] | Prevalence of WPV and associated factors | Primary and secondary care | Nursing, medical, and other personnel | 203 (71%; 29%) | Brazil | Some form of Abuse 40.4% (48%; 22%). |
| 135. | Smed et al., 2020 [152] | Prevalence and characteristics of sexual harassment | Vascular surgery | Faculty of training programs | 149 (22%; 8%) | USA | Sexual harassment (67%;34%) |
| 136. | Speroni et al., 2014 [153] | Incidence of WPV, causes and characteristics | Multiple-hospital system | Nurses | 762 541 (93%; 7%) | USA | Workplace violence 76% (93%; 7%) |
| 137. | Stasenko et al., 2020 [154] | Impact of physician gender and experiencing sexual harassment and gender discrimination | Gynecological Oncology | Gynecologic oncologists | 405 (63%; 36%; Others 1%) | USA | Sexual Harassment (71%; 51%) Discrimination denied of position (33%; 19%) Offensive sexist remarks (58%; 28%) Received a lower evaluation (31%; 14%). |
| 138. | Sun et al., 2017 [155] | Prevalence, frequency and the risk factors for WPV | Tertiary hospitals | Nursing, medical & other personnel | 1899 (61%; 39%) | China | **Physicians:** Physical (11.5%;19.5%) & non-physical (69.6; 74.5%) violence **Nurses:** Physical (12%; 20%) & non-physical (71%; 76%) violence |
| 139. | Tian et al., 2020 [156] | Distribution, types, associated factors for WPV, and its impact | Various Hospitals | Nursing and medical personnel | 3684 (85%; 15%) | China | Emotional abuse (47.3%; 55.4%); Threats (25%; 38%) Physical Abuse (14.5%; 24%) Sexual Abuse (7%; 13%) |
| 140. | Turgut et al., 2021 [157] | Characteristics and causes of violence | Emergency department | Physicians | 157 (37.6%; 62.4) | Turkey | Violence-reported cases (37.6%; 62.4%) |
| 141. | Vargas et al., 2020 [158] | Prevalence and impact of sexual harassment | University Medical School | Faculty members | 705 (48; 52%) | USA | Sexual harassment from insiders (82.5%; 65.1%) Sexual harassment from patients (64.4%; 44.1%) Gender harassment from insiders (82.2%; 64.9%) Gender harassment from patients (64.0%; 44.1%) |
| 142. | Viottini et al., 2020 [159] | 3-year incidence of aggressive acts and risk factors | University Hospital Network | Midwives, nurses and physicians | 364 (77.5%; 22.5) | Italy | Assaults incidences (77.5%; 18.5%) |

(Continued)

**Table 1.** (Continued)

| S. # | Author/s, year | Purpose | Clinical Setting | Professional Category/ies | Sample (Female; Male) | Country/ies | Prevalence of WPV by Type (Female; Male) |
|------|---------------|---------|------------------|--------------------------|----------------------|-------------|------------------------------------------|
| 143. | Vorderwulbecke et al., 2015 [160] | Aggressive and violent incidents, the perpetrator, and the consequences | Primary health care | Primary care physicians | 1500 (40%;60%) | Germany | Aggression (60%; 51%); Sexual harassment (25%; 15%) |
| 144. | Wang et al., 2020 [161] | Gender-based discrimination and bias (GBDB) and their effects | Vascular surgery | Vascular residents | 284 (36%; 64%) | USA | Sexually harassment (25%; 1%) GBDB during training (80%; 14%) Some form of public humiliation (64%, 49%) |
| 145. | Weldehawaryat et al., 2020 [162] | Prevalence of WPV and associated factors | Public health facilities | Nurses | 348 (57%; 43%) | Ethiopia | WPV (61%;39%) |
| 146. | Williams et al., 2021 [163] | Inappropriate behaviors by patients and their effects | Medical residency program | Internal Medicine residents | 33 (41%; 59.3%) | USA | Microaggression by a patient (90.9%; 56.3%) |
| 147. | Xie et al., 2017 [164] | Effects of patient-initiated aggression on quality of life and career | Medical school | Medical students' | 180 (56%; 445) | China | All types of violence (29%; 33%) Sexual harassment (9%; 10%) Physical violence (15%; 3%) |
| 148. | Zachariadou et al., 2018 [165] | Prevalence and forms of workplace bullying | Primary Health care clinics and general hospitals | Medical, Nursing, and other personnel | 167 (71%;29%) | Cyprus | At least one mobbing behavior (49%; 35.7%) |
| 149. | Zampieron et al., 2010 [166] | Perceived aggression, characteristics of aggressors and its type | All levels of health care institution | Nursing personnel | 579 (79%; 21%) | Italy | Aggression (52%; 42%) Verbal aggression (82.8%; 78%) |
| 150. | Zeng et al., 2013 [167] | Violence, the risks and impact of WPV on Quality of life | Psychiatric hospitals | Psychiatric nurses | 392 (77%; 23%) | China | Sexual assault (15.5%; 28.9%) Sexual Physical harassment (15.2%, 18.9%) Physical violence (57.9%; 73.3%) Verbal threat abuse (76.8%, 84.4%) Sexual verbal harassment (22.9%; 31.1%) |
| 151. | Zhu et al., 2018 [168] | Gender differences in WPV | Obstetrics and Gynecology | Physicians | 1425 (87%; 13%) | China | Physical assaults (10.5%; 18.8%) Sexual assaults (1.3%; 5.0%); Verbal abuse (62.2%; 66.7%). |
| 152. | Al Khatib et al., 2023 [169] | Prevalence of physical and verbal violence | Emergency Department | Physicians and nurses | 163 (27%; 73%) | Jordan | Physical (2.3%; 43.7%) Verbal (29.5%; 61.3%) |
| 153. | Al-Wathinani et al., 2023 [170] | Role of healthcare providers, addressing violence and preparedness for managing it | Emergency Department | Physicians, nurses and others | 206 (43.2%; 56.8%) | Saudi Arabia | Physical assault (48.3%; 66.7%) |
| 154. | Alhassan et al., 2023 [171] | Prevalence and circumstances related to physical WPV | Emergency Department | Midwives, nurses, physicians and others | 7398 (48.7%; 51.3%) | Saudi Arabia | Physical attacks (44%; 56%) |
| 155. | AlHassan et al., 2023 [172] | Prevalence of workplace sexual violence and its risk factors | Emergency Department | Midwives, nurses, physicians and others | 7398 (48.7%; 51.3%) | Saudi Arabia | Sexual attack (61%; 39%) |
| 156. | Ashraf et al., 2023 [173] | Gender bias, discrimination and bullying | Medical Schools | Medical students | 377 (65%; 35%) | Pakistan | Sexual assault (8.2%; 6.7%) |

(Continued)

**Table 1.** (Continued)

| S. # | Author/s, year | Purpose | Clinical Setting | Professional Category/ies | Sample (Female; Male) | Country/ies | Prevalence of WPV by Type (Female; Male) |
|---|---|---|---|---|---|---|---|
| 157. | Ayyala et al., 2023 [174] | Gender differences in experienced types of bullying | Internal medicine | Medical residents | 21212 (47%; 53%) | USA | Bullying (harassment) (47%; 53%); Verbal (49%; 51%) Sexual (70%; 30%); Physical (38%; 62%) |
| 158. | Banga et al., 2023 [175] | The nature, risk factors, impact and existing measures for reporting and preventing WPV | Health workers | Nurses, physicians and others | 5405 (53%; 45%; others 2%) | 79 countries | Verbal violence (50.8%; 51%) ; Emotional violence (30.6%; 28%) Sexual violence (7.4%; 3.8%) ; Physical abuse (19%; 24.2%) |
| 159. | Barequet et al., 2023 [176] | Gender-related trends for professional career and personal life performance | Ophthalmology | Physicians | 252 (46%; 54%) | Israel | Any kind of abuse from patients: (79%;77%) Physical Abuse (18%;13%); Sexual harassment (50%;13%) |
| 160. | Bekalu et al., 2023 [177] | Magnitude and associated factors of workplace violence | Public hospitals | Nurses | 534 (45%; 55%) | Northeast Ethiopia | Violence (58%; 42%) |
| 161. | Bekelepi et al., 2023 [178] | Self-reported incidents of physical and verbal violence | Psychiatric units | Nurses | 103 (72%; 28%) | South Africa | Physical assault (74.2%; 25.7%) Verbal abuse (72.2%; 27.7%) |
| 162. | Biurrun-Garrido et al., 2024 [179] | Perceived sexist behavior in their daily life at university and during university teaching | 8-Universities | Nursing students | 317 (86.8%; 12%; others 1.3%) | Catalonia | **During School:** Subjected to intimidatory treatment due to gender expression (28%; 8%) **During Internship:** Subjected to intimidatory treatment due to gender expression (45%; 29%) |
| 163. | Crombie et al., 2024 [180] | Experiences of mistreatment | University | Medical students | 443 (73%; 27%) | South African | Mistreatment (80.9%; 70.8%) |
| 164. | Dawson et al., 2024 [181] | Update rates of assaults | Psychiatry | Psychiatry residents | 275 (64%; 35; others 1%) | USA | Physical assault (18.3%; 29.2%) |
| 165. | Domínguez et al., 2023 [182] | Prevalence and impact of workplace bullying and sexual harassment | Surgery | General surgery residents | 302 (42%; 58%) | Colombia | Occasional bullying (26%; 29%); Continuous bullying (22%;21%) Sexual harassment (29.1%; 4.55) |
| 166. | Ebrahim et al., 2023 [183] | Prevalence of discrimination based on gender | Healthcare workers-Hospital | Nurses, physicians and others | 537 (66%; 34) | Kenya | Discrimination based on gender (27.3%; 20.2%) Physical abuse (5.4%; 3.8%); Verbal abuse (60.9%; 48.9%) |
| 167. | Forsythe et al., 2023 [184] | Experiences of bullying, undermining behaviour, and harassment (BUH) | Vascular diseases physicians | Consultant, fellows, Residents, interns/ students | 587 (35.8%; 62.9%; other 1.5%) | International (28 countries) | Experiences of BUH (53%; 38%) |
| 168. | Grover et al., 2023 [185] | Frequency and types of mistreatments | General surgery and urology | Residents | 23 (35%;65%) | Mid-Atlantic | Mistreatment (88%; 33%), Verbal assault (50%; 33%) |

(Continued)

**Table 1.** (Continued)

| S. # | Author/s, year | Purpose | Clinical Setting | Professional Category/ies | Sample (Female; Male) | Country/ies | Prevalence of WPV by Type (Female; Male) |
|---|---|---|---|---|---|---|---|
| 169. | Ioanidis et al., 2023 [186] | Gender differences in career progression and harassment | Otolaryngology | Residents and attending Physicians | 183 (55.6%; 45.4%) | Canada | **Harassment during residency** (75.2%;37.9%) Verbal (87%; 92%); Sexual harassment (45%; 11%) **Harassment- attending physicians** (68%; 31.6%) Verbal (93%; 65%); Sexual harassment (41%; 22%) |
| 170. | Iqbal et al., 2024 [187] | Relationship between sexual harassment and burnout | Cardiology | Cardiology trainees | 671 (46.5%; 53.5%) | Pakistan | Inappropriate sexual incidents during career (37%; 22%) |
| 171. | Janatolmakan et al., 2023 [188] | Characteristics of physical and verbal violence | Emergency | Hospital nurses | 150 (58.7%; 41.3%) | Iran | Physical Violence (60.2%; 39.8%) Verbal Violence (58.5%; 41.5%) |
| 172. | Lorento Ramos et al., 2023 [189] | Number of workplace aggressions per hospital worker | University Hospital | Nurses, physicians and others | 1118 (72%; 28%) | Canary Island | Physical Aggression (39.7%; 23.2%) Verbal Aggression (71.7%; 60.3%) |
| 173. | Meese et al., 2024 [190] | National trends of violence toward healthcare workers | Healthcare | Healthcare workers, including Nurses | 2659 (60.5%; 39.5%) | USA | Verbal Mistreatment (26.5%; 19.3%) Physical Violence (17.1%; 10.7%) |
| 174. | Parodi et al., 2023 [191] | Gender differences in workplace violence | Health sector | Physicians and nurses | 3056 (57%; 43%) | Latin America | WPV (65.8%; 50.4%); Verbal violence (97%; 97%) Physical Violence (10%; 9%) |
| 175. | Rashid et al., 2023 [192] | To investigate the extent of bullying | Cardiology departments | Junior physicians | 1852 (43%; 57%) | Pakistan | Bullying (13.4%; 10.2%) |
| 176. | Ryan et al., 2023 [193] | Prevalence of patient-initiated discrimination and harassment | Academic orthopedics | Nurses, residents/ fellows, and physicians | 173 (65%; 45%) | USA | Harassment (all staffs): (27%; 24%) Harassment (residents and faculty) (46%; 24%) |
| 177. | Santosa et al., 2023 [194] | Incivility experiences, attributes and associated perpetrators | Academic surgery programs | Faculty and residents | Residents/ fellows: 143 (58%; 42%) Faculty: 183 (41%; 59%) | USA | Incivility experiences among surgeons (77%; 6%) |
| 178. | Shahjalal et al., 2023 [195] | Prevalence and associated factors of physical violence | Tertiary care hospitals | Physicians | 406 (49%; 51%) | Bangladesh | Physical Violence (36%; 64%) |
| 179. | Sobel et al, 2023 [196] | Differential experiences of harassment in pre-residency interviews | Graduate medical education | Orthopaedic surgery Residents | 94 (16%; 83; no response 1%) | USA | Harassment (20%; 0%) |
| 180. | Tavolacci et al., 2023 [197] | Prevalence and factors associated to gender-based violence | Health Campus at Rouen and nursing school | Students, including midwifery, nursing and medicine | 1152 (82.6%; 17%; others: 0.4%) | France | GBV (93.7%; 5.4%) |
| 181. | Veronesi et al., 2023 [198] | Introducing a systematic WPV surveillance and underreporting system-before and after its implementation | Two public referral hospitals | Nurses, physicians and others | 7982 (74.7%; 25.3%) | Italy | Before complete data was not available to the researchers. **After implementation:** WPV (65.7%; 34.3%) Physical violence (47.7%; 67.2%); Verbal (92.3%; 88.1%) |

(Continued)

**Table 1.** (Continued)

| S. # | Author/s, year | Purpose | Clinical Setting | Professional Category/ies | Sample (Female; Male) | Country/ies | Prevalence of WPV by Type (Female; Male) |
|---|---|---|---|---|---|---|---|
| 182. | Vu et al., 2023 [199] | Workplace violence and its variables affect acts of violence | University Students | Medical, nursing and others | 550 (75%; 25%) | Vietnam | Any type (29.6%; 37.7%); Physical violence (10%; 22.5%) Verbal Violence 25.7%; 29.7%); Sexual violence (4.6%; 5.8%) |
| 183. | Yang et al., 2023 [200] | Associations between workplace violence and patient safety behaviours | General Hospitals | Nursing interns | 466 (83.5%; 16.5%) | China | Verbal Abuse (73.5%; 75.3%); Threaten (95.6%; 89.6%) Physical violence (98.5%; 98.7%); Sexual Harassment (97%; 100%) Sexual assault (99%; 99%) |
| 184. | Nam et al., 2023 [201] | Experiences of clinicians with patient-perpetrated sexual harassment | Departments of Urology and Obgyn at the University | Clinicians | 128 (76%; 24%) | Michigan | **Urology:** Unwanted sexual attention (68.8%; 22.7%) Gender harassment (84.4%; 40.9%); Sexual Coercion (15.6%; 0%) **OBGYN**; Unwanted sexual attention (68.8%; 54.6%) Gender harassment (84.4%; 68.2%); Sexual Coercion (3.1%; 0%) |
| 185. | Yan et al., 2023 [202] | National prevalence of WPV and associated factors | Emergency | Physicians | 14848 (29.5%; 70.5%) | China | Any Type of violence (87.7%; 91.6%) Physical (37%; 57.5%); Non-physical (87.2%; 91.2%) |

and women); to report data we defined *gender* as a binary (male/female) for this review. The specific objectives were:

1. Map the most frequent forms and prevalence of GB-WPV for midwives, nurses, and physicians in different contexts and clinical settings.

2. Identify the gendered dimensions of the health workforce that underpin violence against male or female health workers.

3. Identify gaps in the state of knowledge to recommend empirical research studies.

## Methods

### Protocol registration and study design

We conducted the scoping review according to Joanna Briggs Institute's (JBI) revised guidelines [22]. The protocol (S2 Appendix) was registered on the Open Science Framework on January 14, 2022, and can be accessed at https://osf.io/t4pfb/. We utilized the scoping literature review design to address the questions and to cater to the heterogeneous and complex literature because it is an appropriate method to explore the extent of the literature, map and summarize the evidence, and identify and analyze the knowledge gap to inform future research [23].

This framework consists of eight steps and originated from the seminal framework of Arksey and O'Malley's scoping review [24], which was advanced by Levac and colleagues [25]. In the revised guidelines, JBI aligned the eight steps with the Preferred Reporting Items for Systematic and Meta-Analyses extension for Scoping Reviews (PRISMA-ScR) [26], which is used to report the conduct of the scoping review that provided rigour, transparency, and trustworthiness [23]. Please see the filled PRISMA-ScR Checklist S3 Appendix. The first step of the scoping review framework is to align research objectives with the title and the inclusion criteria, which we have described in the earlier section and the inclusion criteria (see Box 1).

> ## Box 1. Study selection criteria
>
> Inclusion Criteria for Studies
>
> 1. The study participants included midwives, nurses, and/ or physicians who experienced WPV during their careers.
>
> 2. Provided sex-segregated data for any form of violence among midwives, nurses, and physicians, including students, globally.
>
> 3. Published in English and after 2010.
>
> **Exclusion criteria**
>
> 4. Studies that did not provide sex-segregated data
>
> 5. Exclude qualitative studies, systematic/ scoping reviews, concept or theoretical papers, and theses.

## Search strategy

The research team developed a comprehensive search strategy in consultation with the health sciences librarian. The search focused on the systematic search of published literature in the databases, including Ovid MEDLINE: Epub Ahead of Print, In-Process and Other Non-Indexed Citations; this search was then translated into CINAHL Plus, APA PsycINFO, Web of Science, Gender Studies Database, Applied Social Sciences Index & Abstracts (ASSIA) and Sociological Abstracts (see Ovid MEDLINE search strategy in S4 Appendix).The search terms related to the population (midwifery, nursing, and physicians), concepts (violence and gender-based violence), and the context (healthcare) and appropriate combinations were used for searching for the scoping review [23]. These terms were identified from the preliminary literature search on different aspects of WPV using Google Scholar. The final search results were exported to EndNote. After de-duplication, these sources were imported into Covidence, an online program to streamline the screening process by two independent reviewers. To cover multifaceted gender-based WPV comprehensively from global perspectives required significant conceptual development and synthesis. The most recent search of the literature review for this study was conducted on 11 February 2024.

## Evidence screening and selection

The identified sources were selected based on the set inclusion criteria in Box 1. Two independent reviewers screened the title and abstracts for shortlisted sources. The discrepancies were resolved with discussion and consensus and by reviewing the complete source, followed by a full-text review for selected sources against the set inclusion criteria by two reviewers and the

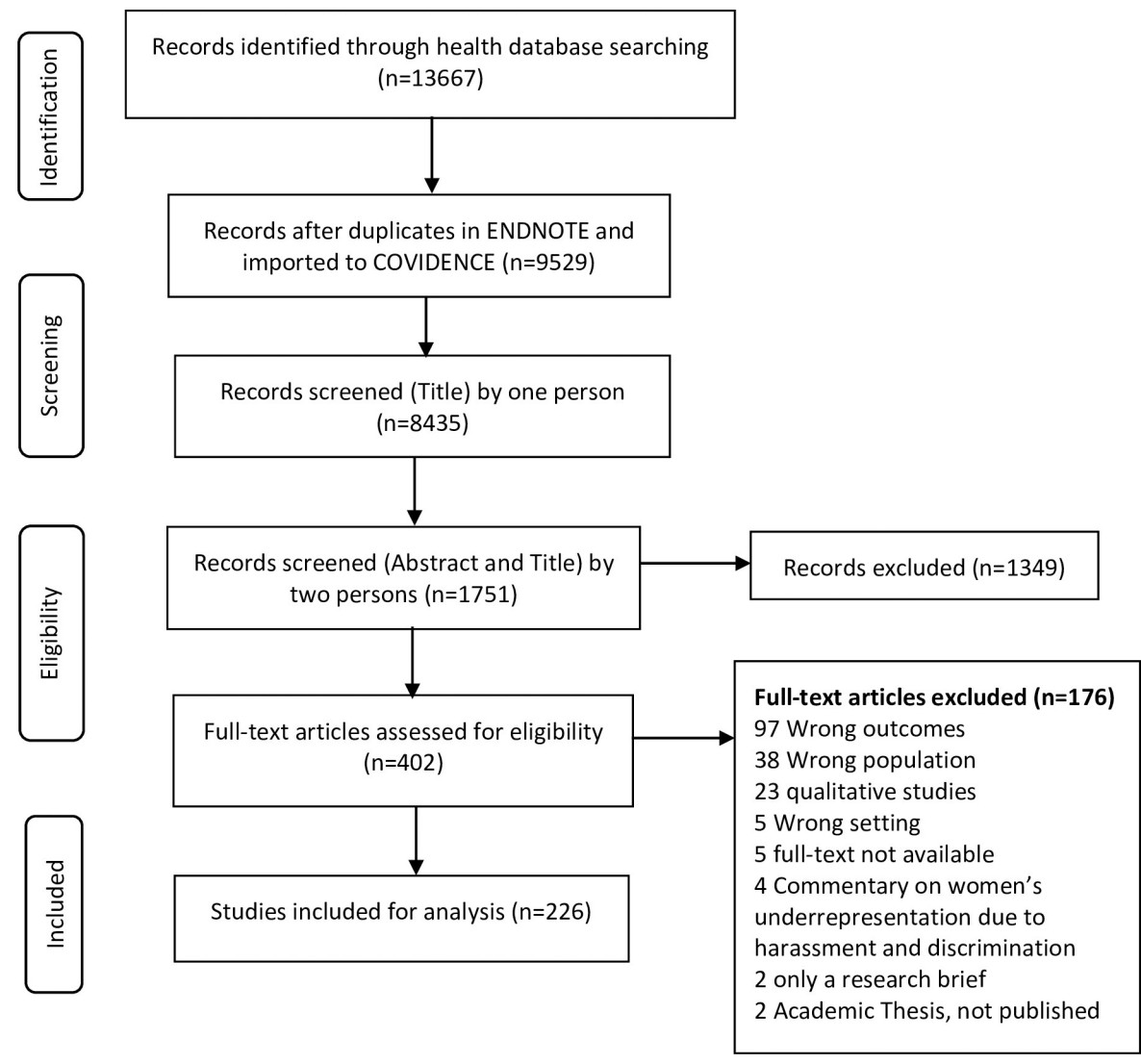

**Fig 1. PRISMA flow diagram for screening and selection.**

abstraction of the information independently. The selection process is presented in the PRISMA diagram (Fig 1).

## Data extraction

Data from all included studies were extracted into Microsoft Excel; information was charted regarding author/s and year, title, source, country, objective/purpose, study design and methods of analysis, sample size, and category of health worker and sex/gender-segregated, key findings, and research gaps indicated by the author/s [23].

## Data analysis and synthesis of results

Data from articles that reported descriptive statistics regarding sex/gender were included in mapping the prevalence of GB-WPV (Table 1) for several types/forms of WPV and the clinical setting. Some studies treated gender as a risk factor or a predictor for the outcomes of WPV, harassment, and discrimination using inferential statistical analysis presented in Table 2. We

**Table 2. Inferential statistics, presenting gender as a factor/predictor for various outcomes of WPV across clinical settings, professions and countries.**

| S. # | Author/s, year | Purpose | Clinical Setting | Professional Category/ies | Sample (Female; Male) | Country/ies | WPV and its association/outcomes by gender (Female; Male) |
|---|---|---|---|---|---|---|---|
| 1. | Rodriguez-Acosta et al., 2010 [203] | Injuries resulting from physical assault and associated risk factors | University Hospitals | Nursing staff | 220 injuries (86%; 14%) | Duke, North Carolina, USA | While the number of assaults was greater among women than men, their risk was lower (RR = 0.70). Nursing aids had a greater risk of physical assault than nurses (RR = 1.51). |
| 2. | Gan-young et al., 2018 [204] | Prevalence and factors associated with WPV | The grassroots communities | General practitioners | 1015 (35%; 65%) | China | Male GPs were more likely to have experienced physical violence (OR = 3.4, CI 2.1–5.6) and non-physical violence (OR = 2.6, CI 1.7–3.0) than female GPs. |
| 3. | Hahn et al., 2013 [205] | Factors associated with patient and visitor violence | University general hospital settings | Physicians, nurses and midwives | 2495 (82%; 18%) | Switzerland | Gender factored WPV (no clear description provided) |
| 4. | Hu et al., 2022 [206] | Relationship between a medical specialty's gender composition and harassment | Association of American Medical Colleges | Physicians | 6000 (29%; 66%); non-binary | USA | Greater representation of women within a specialty is associated with a lower prevalence of harassment experienced by men and women physicians (e.g., threats of physical harm for women (OR = 0.973, CI 0.954–0.992) and men (0.984, CI 0.974–0.993) and unwanted sexual advances for women (OR = 0.976, CI 0.967–0.984) and men (0.988, CI 0.981–0.995). |
| 5. | Jacobson et al., 2022 [207] | Identify sources of Horizontal Violence using the Negative Acts Questionnaire-Revised | Emergency medicine | Medical residents | 23 (56%; 44%) | USA | Horizontal violence (1.3; 1.1, P = .01). By category, women indicated a higher incidence of work-related violence. |
| 6. | Fu el al., 2021 [208] | Level of fear of future WPV and influencing factors | Tertiary hospitals | Nurses | 1898 (94%; 6%) | China | Scores for fear of future WPV was significantly higher among women (p < 0.001) |
| 7. | Lund et al., 2022 [209] | Relationship between gender, gendered microaggressions, and burnout | 7 academic institutions. | Surgical faculty | 111 (40%; 60%) | USA | Women reported higher levels of gendered microaggressions (p = 0.001), which predicted higher levels of burnout (p = 0.001). |
| 8. | Wang et al., 2022 [210] | Relationship between burnout, organizational commitment and turnover intention, and factors related to WPV | ICU | Nurses | 305 (68%; 32%) | China | There were increased odds of experiencing WPV among nurses with lower professional titles, male nurses (OR = 2.7, CI = 1.310 to 5.944), and those with less than five years of experience. |
| 9. | Wright & Khatri, 2015 [211] | Relationship between bullying and its outcomes (psychological/behavioral responses) | Teaching hospital network | Nurses | 1078 (91%; 9%) | USA | Male nurses experienced significantly higher work-related bullying than female nurses p < .067). No significant differences in person-related bullying, which had a significant positive relationship with both psychological/behavioral responses and medical errors. |
| 10. | Bambi et al., 2014 [212] | Lateral Hostilities (LH) and its effects on the quality of life | Prehospital EMS, emergency department, ICU, and OR | Nurses | 1202 (61.5%; 38.5%) | Italy | Desire to leave the nursing profession because of the LH was (15.5% and 9%); however, gender was not statistically significant for LH. |

(*Continued*)

**Table 2.** (Continued)

| S. # | Author/s, year | Purpose | Clinical Setting | Professional Category/ies | Sample (Female; Male) | Country/ies | WPV and its association/outcomes by gender (Female; Male) |
|---|---|---|---|---|---|---|---|
| 11. | Tekin and Bulut, 2014 [213] | Verbal, physical, and sexual abuse status | Operating room | Nurses | 360 (92%; 8%) | Turkey | A significant relationship- between gender and educational status (p<0.05); women were more exposed to verbal abuse. |
| 12. | Koukia et al., 2014 [214] | Prevalence of WPV | General Hospital | Healthcare staff, including nursing and medical personnel | 250 (74%;26%) | Greece | Women were more likely to experience sexual (p<0.012) and physical violence (p<0.014). |
| 13. | Yang & Zhou 2021 [215] | Prevalence, severity, and the risk factors contributing to WPV | Operating room | Nurses | 471 (91%; 9%) | China | The mean score for bullying (1.93; 1.59); Gender was a (p < .001) significant determinant of bullying; Men were 0.373 points less likely to be bullied than women. |
| 14. | Ode et al., 2021 [216] | The extent of perceived occupational opportunity and workplace discrimination | Surgery | Black orthopedic surgeons | 274 (21%; 78.5%); 01 declined | USA | Microaggressions range for women (66% to 98%) and men (53% to 87%) across all four solicited questions. |
| 15. | Favaro et al, 2021[217] | Relationships among sex, empowerment, workplace bullying and job turnover intention | New graduate nurses from 10 provinces | New graduate nurses | 1008 (92.5%; 7.5%) | Canada | Male nurses (M = 1.778, SD = 0.86) reported significantly (= p < .001) more frequent workplace bullying than female nurses (M = 1.487, SD = 0.65). No significant difference by sex for either structural empowerment or job turnover intention |
| 16. | Obeidat et al., 2018 [218] | Workplace bullying and its relationship with perceived competence | Private hospitals | Registered nurses | 274 (49%; 51%) | Jordan | Men were more likely to report a higher overall rate of perceived workplace bullying (p < 0.001) than women. The perceived competence score among severe bullying was significantly lower (<0.001). |
| 17. | Alameddin et al., 2015 [219] | Prevalence, characteristics, and consequences of WPV | Database of the Order of Nurses in Lebanon | Nurses | 593 (79%; 21) | Lebanon | Male nurses had 2.22 times the odds of exposure to physical violence compared to females (95% CI 1.14–4.35, p- 0.019). |
| 18. | al-Omari, 2015 [220] | Prevalence of WPV and associated factors | 11 General hospitals | Nurses | 468 (47%; 53%) | Jordan | Female nurses were 0.5 times less likely to report being physically attacked than male nurses (p = 0.003). Female nurses were 1.5 times more likely to report being verbally abused than male nurses (p = 0.046). |
| 19. | Campbell et al., 2011 [221] | Prevalence of WPV and demographic | 4 health care institutions | Nursing personnel (all categories) | 2166 (91.5%; 8.5%) | USA | Males were nearly twice as likely to have experienced physical WPV compared to females. |
| 20. | Esmaeilpour et al., 2011 [222] | Frequency and nature of WPV | Emergency department | Nurses | 186 (89%; 11%) | Iran | Male nurses were the victims of physical violence more often than female nurses (p = 0.000). |
| 21. | James et al., 2011 [223] | Attitudes toward and perception of the prevalence of aggression | Psychiatric in-patient settings | Mental health nurses | 76 (71%; 31) | Nigeria | Male nurses reported significantly higher episodes of aggressive spitting behaviour (p<0.011) as well as physical violence (p<0.010). |

*(Continued)*

**Table 2.** (Continued)

| S. # | Author/s, year | Purpose | Clinical Setting | Professional Category/ies | Sample (Female; Male) | Country/ies | WPV and its association/outcomes by gender (Female; Male) |
|---|---|---|---|---|---|---|---|
| 22. | Joa and Morken, 2012 [224] | WPV prevalence, its causes and associated factors | Out-of-Hours primary care centres | Physicians, nurses, and others | 536 (70%; 30%) | Norway | Men were more at risk of physical abuse (OR = 2.36, CI 1.11–5.05) and verbal abuse (OR = 1.23, 0.68–2.18). |
| 23. | Han et al., 2022 [225] | WPV and its association with workforce stability and well-being | Psychiatry | Psychiatrists and nurses | 14264 (75%; 25%) | China | Males were 1.75 (95% CI = 1.53, 2.00) times more likely to report encountering violence than females. |
| 24. | Serafin & Czarkowska-Pączek, 2019 [226] | Prevalence, the most common negative acts, and the risk factors of bullying. | Polish healthcare facilities | Nursing personnel | 411 (96%; 4%) | Poland | Women were more often affected by 'being humiliated or ridiculed in connection with their work' (p = 0.040), 'being ordered to do work below their level of competence' (p = 0.010), and 'having key areas of responsibility removed or replaced with more unpleasant tasks' (p = 0.005). |
| 25. | Al-Ghabeesh & Qattom, 2019 [227] | Prevalence of bullying and the impact of preventive measures on productivity | Emergency department | Nurses | 120 (35%; 65%) | Jordan | No significant differences based on the gender of the participant (p = 0.07). |
| 26. | Ceballos et al., 2020 [228] | Characteristics, related factors, and consequences of WPV | Emergency care unit | Nurses | 80 Reported majority were females | Brazil | Female nurses suffered verbal violence 5.83 times higher than men (OR = 5.83; p = 0.026). |
| 27. | Sellers et al., 2012 [229] | Degree of horizontal violence (HV) | 19 Healthcare organizations | Registered nurses | 2659 (93%; 7%) | New York State, USA | Women reported significantly greater (p < .05) knowledge of and being a victim of HV than men. |
| 28. | Kelly et al., 2015 [230] | Prevalence and the relationships between static and dynamic staff risk factors for patient-on-staff assault | Forensic hospital | Overall staff | 488 (69%; 31%) | California, USA | Men experienced higher scaled frequencies of assault than women (4-point Likert scale, 0–3] mean = (0.46 vs 0.33, p = 0.02). |
| 29. | Vezyridis et al., 2015 [231] | Prevalence, characteristics, factors, and suggestions for improving WPV | emergency departments | Nursing and medical personnel and a few others | 220 (62%, 38%) | Cyprus Republic | No significant differences between the participant's gender. |
| 30. | Fafliora et al., 2015 [232] | Prevalence and the characteristics of WPV | Primary, secondary, and tertiary care hospital | Nurses | 80 (83%;17% | Greece | Men (OR, 0.08, CI 0.01–0.56) and higher experience nurses (OR, 0.82, CI 0.70–0.097) were less affected by WPV. |
| 31. | Askew et al., 2012 [233] | Perceived workplace bullying and consequences | All doctors registered with the Australian Medical Registration Board | Doctors from various department | 747 (53%; 47%) | Australia | There were no differences in the prevalence of bullying between the sexes. Victims of bullying had poorer mental health (p<0.001) |
| 32. | Lindquist et al., 2020 [234] | WPV and characteristics associated with increased risk | National emergency medicine conference in Myanmar | Physicians and medical students | 63 (56%; 41%; Missing: 3%) | National emergency medicine conference in Myanmar | Women were more likely to experience verbal assault (OR = 1.18, 0.42–3.33). |
| 33. | Yohe et al., 2020 [235] | Workplace hazards, including the rate of WPV during training. | Orthopedic residency | Orthopedic residents | 1207 (17%; 83%) | USA | Gender was not statistically significant (OR = 1.07, 0.63–1.84, p-0.79). |
| 34. | Firenze et al., 2020 [236] | The prevalence and perpetrators of WPV | Hospitals | Medical personnel | 4545 (57%; 43%) | Italy | Males experience almost three times higher (aOR 2.09, 95% CI 1.51–2.88, p<0.001) |
| 35. | Aghajanloo et al., 2011 [237] | Extent and types of violence during clinical training | Iran Medical University | Nursing students | 180 (72%; 28) | Iran | No significant relation between students' sex and the frequency of insult (p = 0.051) |

(Continued)

**Table 2.** (Continued)

| S. # | Author/s, year | Purpose | Clinical Setting | Professional Category/ies | Sample (Female; Male) | Country/ies | WPV and its association/outcomes by gender (Female; Male) |
|---|---|---|---|---|---|---|---|
| 36. | Farid et al., 2021 [238] | Experiences of discrimination and microaggressions, | Academic obstetrics and gynecology | Physicians | 87 (75%; 25%) | USA | Most physicians (71%) who had ever experienced discrimination attributed these experiences primarily to their gender. |
| 37. | Jaradat et al., 2018 [239] | Associations between workplace aggression (WPA) and psychosomatic symptoms | Hospitals and primary care clinics | Nurses | 341 (62%; 38) | Palestine | There were no significant differences between sex and workplace aggression resulting in psychosomatic symptoms (raged from 0.04–0.09). |
| 38. | Periyakoil et al., 2020 [240] | Microaggressions and demographic characteristics that affect the reporting | Medical faculty members | Medical personnel | 124 (64%; 36%) | USA | Women reported much higher frequencies of the microaggressions depicted in 33 of 34 microaggression videos (p-value raged <0.001–0.042) |
| 39. | Vingers, 2018 [241] | Gender differences for bullying | Nursing Education | Nursing students | 107 (87%; 13%) | USA | There was no significant difference in the frequencies of reported bullying behaviors for male and female nursing students (< .05 level). |
| 40. | DiFiori et al., 2023 [242] | Prevalence and nature of bullying | Orthopaedic surgery | Surgeons, fellow and residents | 105 (12%; 84%; others 6%) | USA | Demographic information including sex did not have a statistically significant (p-0.167) association with self-reported bullying |
| 41. | Meng et al., 2023 [243] | Occurrence and correlated factors of physical and verbal violence | Emergency | Physicians | 10457 (27%; 73%) | China | Men had a higher risk of physical violence (aOR = 2.45; 95% CI = 2.17–2.76) and verbal violence (aOR = 1.70; 95% CI = 1.51–1.92) than women. |

could not calculate a mean score for violence based on gender because of variability in the definition of the terms and the concepts from various contexts (Please see S1 Appendix for definitions of the various forms of WPV). Besides prevalence and gender as a risk factor, other data are described qualitatively, including the risk factors or predictors, the distribution of WPV based on professional category, the professional hierarchy, the perpetrators, reporting systems, and any preventive interventions and their outcomes.

Given the overall objective of the review to map the most frequent forms and prevalence of GB-WPV for midwives, nurses, and physicians in different contexts and clinical settings, a quality assessment of the identified sources was not conducted. This paper describes the prevalence of WPV based on gender, the influence of gender and the clinical setting, the professional status/role, and the professional hierarchy and gendered roles and responsibilities that have marginalized either males or females within the professional categories. Other aspects, such as perpetrators of GB-WPV and findings of qualitative studies, will be reported elsewhere.

# Results

## Description of identified studies

After de-duplication, 9529 possible references were imported for screening in the Covidence. These studies were screened against the title by one person, 1751 were shortlisted to be

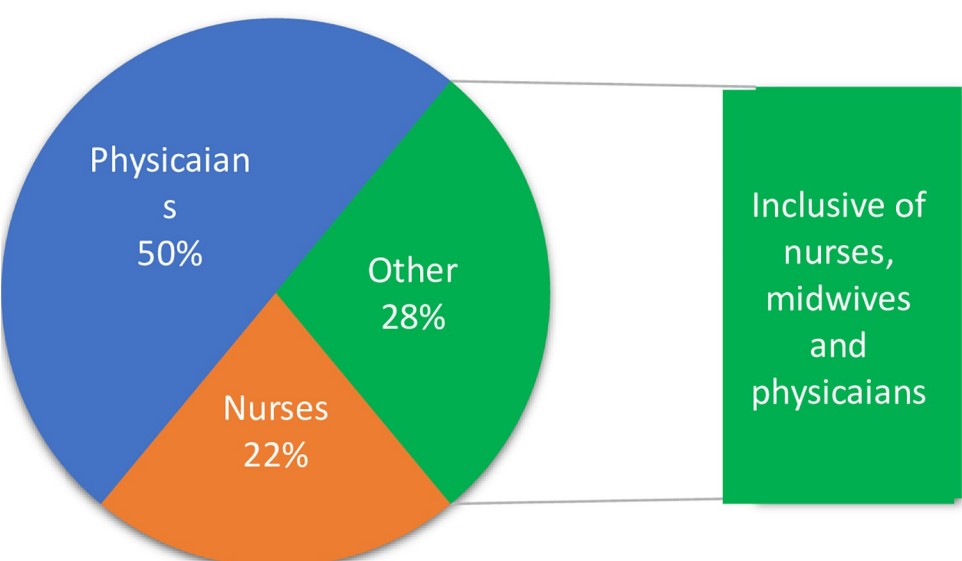

**Fig 2. Percentage of studies, sampled professionals.**

screened (for title & abstract) by two independent reviewers, and 402 were assessed for full-text eligibility. After applying the inclusion and exclusion criteria, 226 studies were retained and analyzed to report on GB-WPV (PRISMA diagram, Fig 1). We included studies published between 2010–2024.

## Study design and population

Fig 2 presents the proportion of studies (226) included in this paper that used a sample of nurses, physicians and/or the entire workforce.

We present findings for midwifery, nursing, and medical workforce samples and subsamples only from across the world. Most of the included studies were conducted in the USA (n = 63), followed by China (n = 20), Turkey (n = 9), Australia (n = 9 [2 studies also included New Zealand]), Italy (n = 9), Saudi Arabia (n = 9), Iran (n = 8), The UK (n = 6), Canada (n = 6), Jordan (n = 6), Brazil (n = 5), Greece (n = 5), Pakistan (n = 5), Ethiopia (n = 3), India (n = 3), Norway (n = 3); 11 countries/special regions including Bangladesh, Taiwan, Ghana, Spain, Germany, Kenya, Poland, South Africa, Switzerland, Palestine, and Cyprus had two studies each (n = 22); 29 other studies conducted one in each country and Six used a global sample from other platforms, such as conferences and professional/research forums.

## Prevalence of gender-base workplace violence

A total of 226 studies provided sex-segregated descriptive data for WPV. Of the 226, 185 studies [5, 7, 11–14, 19–21, 27–202] provided sex-segregated prevalence data (descriptive) for different forms of WPV (see Table 1). Forty-one studies [203–243] provided inferential statistics for violence about gender as a risk factor or predictor of consequences of WPV (Table 2). Of 185 studies that provided descriptive statistics for various forms of WPV, 119 studies (64%) reported a higher prevalence for women participants for all forms of violence as opposed to 31 studies (17%) that reported a higher prevalence for men participants (Fig 3).

Furthermore, 35 studies (19%) reported a higher prevalence of various forms of violence for either men or women, such as in India, where physical violence was higher for men (16%)

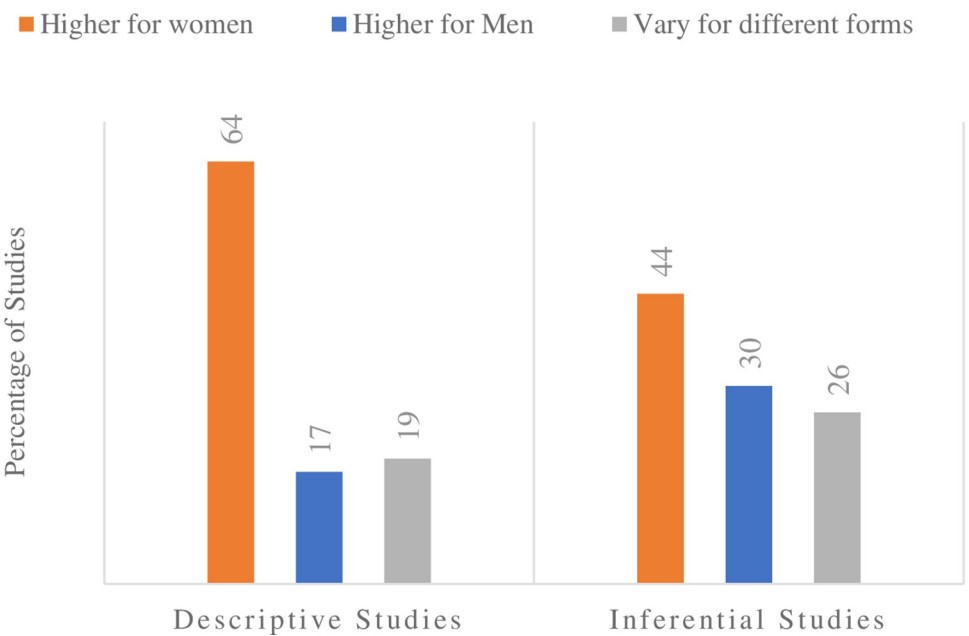

**Fig 3. Proportion of studies presented gender segregated WPV data.**

than women (4%). In contrast, threats were higher for women (58%) than men (47%) [11]. Similarly, the prevalence of physical violence was higher for men compared to women in Australia [83], USA [174], China [5, 89, 155], Iran [115, 129], Iraq [123], Italy [198], Austria [150], and Pakistan [133]. On the other hand, non-physical violence was higher for females in Australia [83], China [5, 89], Canada [95], Iran [115, 129], Iraq [123], USA [174], and Austria [150]. A recent world-wide study collected responses (5405) from all healthcare professionals, including nurses and physicians from 79 countries reported on nature, risk factors and impact of violence [175]. Banga and colleagues (2023) reported higher emotional violence (30.6% Vs. 26%) and sexual violence (7.4% Vs.3.8%) among women and higher physical violence (24% Vs. 19%) among men. In the same study, being nurse had higher odds (OR = 1.95; 95% CI 1.46–2.59) of violence than physicians (OR = 1.70, 95% CI 1.33 to 2.18) and other providers (OR = 1.58; 95% CI 1.21 to 2.05).

## Factors affecting workplace violence

Not all studies were aimed at assessing workplace violence based on gender; therefore, we included all studies that either provided data on gender-based workplace violence or findings that indicated gender was a factor for WPV or a predictor for various outcomes of WPV. The latter group of studies had various aims, including: to assess factors associated with workplace violence [19, 27–33, 139, 204–207, 243], to assess the association between aggression, psychological distress, and job satisfaction [34, 35, 120, 187], to determine injuries resulting from physical assaults [203], to determine the relationship between WPV and psychological and behavioral responses [35–37, 161, 200, 202, 208–211], to assess effects of lateral violence and its consequences [38, 212], and to assess the preparedness to respond to sexual harassment before and after a workshop [39, 65]. In addition to gender as the basis for WPV, we classified other factors associated with WPV that were not explicitly gendered, such as age and status (presented below). However, these factors are implicitly aligned with the broader societal norms and the inherent patriarchal structure of the health system, including the health

workforce that assigns roles and responsibilities to male and female healthcare providers based on their gender, leading to differential experiences, including exposure to workplace violence, we present in the following section.

**Various forms of violence and gender differentials.** Gender influenced both female and male healthcare providers' experiences of different forms of workplace violence. Women were the most targeted for non-physical violence, including verbal violence [5, 7, 13, 40, 41, 80, 83, 92, 98–100, 103, 106, 108, 110, 114, 115, 117, 123, 125, 129, 133, 135, 147, 150, 166, 175, 178, 183, 185, 188–190, 198, 213], sexual harassment [7, 12, 20, 21, 40, 74, 75, 78, 79, 81, 82, 85, 86, 88, 93, 94, 99, 100, 104, 108–110, 116–119, 121, 126–129, 131, 132, 143, 148–150, 152, 154, 158, 160, 161, 172, 173, 175, 176, 179, 182, 186, 193, 196, 201, 213, 214], bullying [7, 42, 73, 87, 94, 99, 101, 119, 121, 132, 135, 144, 182, 184, 192, 215], and discrimination [75, 79, 81, 104, 107, 108, 117–119, 121, 122, 128, 131, 132, 149, 183, 216]. However, women also experienced physical violence [7, 99, 106, 110, 114, 124, 125, 136, 142, 148, 164, 176, 178, 183, 188–191, 214], threats [11, 41, 93, 200], and aggression [90, 113, 166] from various sources. Men also experienced non-physical violence, including verbal [11, 12, 70, 72, 97, 112, 115, 139, 142, 146, 167–169, 174, 186, 199, 200, 202, 243], sexual harassment [115, 156, 164, 167, 186, 199, 200], bullying [43, 120, 129, 145, 182, 211, 217, 218], and discrimination [95]. Physical violence was the only form of violence with a higher reported prevalence in men compared to women [11, 12, 29, 31, 70, 71, 83, 89, 97, 103, 112, 115, 123, 128, 129, 133, 146, 155, 156, 166–171, 174, 175, 181, 195, 198, 202, 204, 219–224, 243].

The studies also reported on the differential effects of WPV on men and women. For instance, women were significantly more likely than men to experience changes in mental health and social behaviours [89, 96, 117, 130, 191, 200, 208] because of violence. Violence is also reported to have affected female healthcare providers' career goals and development [44, 88, 118, 154, 175], leading to dissatisfaction [34, 118, 175] and burnout [34, 45, 102, 117, 121, 128, 174, 209], and leaving or considering leaving the workplace [7, 46, 80, 175, 212, 217, 219, 225]. However, these consequences were rarely highlighted for men. A recent study compared and found an inverse association between workplace violence and patient safety behaviours among nursing interns in China, in that male nursing interns were more likely to exhibit poor patient care behaviour. In contrast, female nursing interns were more likely to exhibit poor mood [200]. In addition to gender, the following demographic factors were related to the risk of WPV among healthcare providers.

**Age.** Younger age was found to be a significant risk factor for violence by several studies for both male and female healthcare providers [7, 12, 14, 33, 35, 36, 47–50, 73, 76, 78, 83, 89, 91, 92, 96, 98, 103, 123, 139, 141, 147, 152, 165, 190, 195, 198, 219, 224]. The studies that reported a higher prevalence of WPV in general for women also reported that the younger age of the health care provider was associated with a higher risk of violence in various contexts, including verbal abuse [14, 36, 83, 92, 98, 123, 204, 219], bullying, or mobbing [14, 47, 50, 73, 165, 226], sexual harassment [12, 36, 44, 49, 78, 152, 224] and aggression [147, 159]. A few studies reported a higher prevalence of violence among males, especially of a younger age, who were at increased risk of verbal violence [139] and bullying among nurses [211, 218] and physical violence among physicians, nurses, and others [195, 224]. Some studies found that advanced age was a protective factor that decreased the risk of WPV [36, 89, 91, 191].

**Work experience.** Several studies reported on an association between the years of experience and the occurrence of different forms of WPV, and most studies found this relationship to be inversely proportional [12, 35, 49, 76, 98, 99, 111, 123, 124, 146, 152, 171, 194, 203, 204, 212, 214, 219, 223, 227–229]. We analyzed the forms of violence which had a higher prevalence for either men or women to examine any connection to years of work experience. Studies reported that men experiencing more verbal abuse [219], physical violence [123, 146, 171],

and bullying [218] had the least work experience. Similarly, less experience as a factor for verbal abuse [98], sexual harassment [12, 149, 152], physical violence [124], horizontal violence [229], bullying [35] and aggression [159], in studies which reported women as victims of WPV. In addition, the increased experience seemed to be a protective factor against physical violence among nurses [228].

**Professional and organizational hierarchy.** The professional hierarchy among midwives, nurses, physicians, and others [13, 47,51, 83, 87, 90, 96, 105, 110, 134, 140, 143, 159, 165, 169–172, 183, 189, 198, 205, 224] and the organizational hierarchy within or between different professionals [47, 80, 95, 116, 152, 165, 184, 194, 204, 226, 230, 242] was found to contribute to various forms of workplace violence. Some studies reported a higher prevalence of WPV among women across the workforce; they reported that female nurses' experience of WPV was higher than female physicians in Australia [83], Brazil [96], Canary Island [189], France [197], Kenya [183], Italy [31, 105], Japan [110], Turkey [138], Ghana [36], and India [146]. Verbal abuse was prevalent for female nurses in the Caribbean [13], Norway [224], and Cyprus [231] from patients or relatives; aggression from staff and colleagues was prevalent in England [90], from patients and visitors in Switzerland [205], and from patients in Poland [51], Israel [176] and Italy [159]; mobbing was found to be prevalent for female nurses in Turkey [140] and Cyprus [165], and sexual harassment was found to be prevalent among female nurses in Nepal [143] and Saudi Arabia [172]. A smaller number of studies reported a higher prevalence of violence for males in Colorado, USA [134], Turkey [141], China [155], Jordan [169] and Saudi Arabia [170, 171]. The last three studies in Jordan and Saudi Arabia reported more physical violence for men in emergency departments, where most assailants were patients and their relatives [169–171], which could be explained by overcrowding, increased waiting time, and increased workloads for health care providers [171].

In a Greek study, the prevalence of bullying was higher among women (nurses and physicians); however, women physicians self-labelled as being victims more often than nurses [87]. Another study in India reported that physician participants experienced more episodes of WPV (77%) followed by nurses (48%); the overall prevalence rates were higher for women [14]. In the context of Saudi Arabia, in a study, both physicians and nurses (68.5% vs 68%) [73], and in another study, nurses (82%) and medical residents (15%) experienced bullying from various sources [35]; both these studies found overall higher prevalence for women than men (66% vs.49%) [73], (69% vs. 31) [35], that explain gender intersection with professional hierarchy that victimized women.

The formal organizational hierarchy played a role in perpetuating workplace violence due to the inherent power dynamics in the health workforce. One study in Cyprus [165] reported that mobbing, which involves hostile and unethical communication directed systematically towards an individual, was prevalent among women in the workforce, with nurses significantly more affected than physicians. The same study also reported that chief and senior nurses were significantly less exposed to bullying behaviours than junior nurses (33.3% and 46.7% vs. 56.1%). Similarly, a study in a forensic hospital in California, USA, reported men experiencing a higher frequency of assaults in wards than men in clinics and supervisory positions [230]. Banga and colleagues [175] included participants from 79 countries who reported that 16% of the participants experienced aggression from their supervisors. In the study, nurses were more likely to experience higher levels of violence than physicians. Fifty-five percent of victims reported job dissatisfaction, and 25% were willing to quit.

Several studies reported exclusively on nursing personnel. A study conducted in Poland reported seniority as a protective factor against bullying and that nurse managers experienced a significantly lower level of bullying compared with clinical nurses and nurse coordinators [226]. In China, male nurses who had lower professional titles in intensive care units had

higher odds of experiencing WPV [210]. Another study from Lebanon reported male nurses' higher risk for exposure to violence, and managers/supervisors were found to be the most common perpetrators of verbal abuse [219]. In this Lebanese study, male nursing students reported experiencing discrimination in the female-dominated profession.

Interestingly male medical professionals likewise reported violence and discrimination based on gender. In fact, several studies reported a significant relationship between the role, seniority, and the experience of violence among medical personnel. For instance, male medical residents and General Practitioners (GPs) had higher odds of violence than specialists in an emergency department in Turkey [76], GPs in China [204], and junior residents in India [146]. On the other hand, female medical personnel experienced more harassment and discrimination throughout their career, including in academia, regardless of role or seniority. Of 185 descriptive studies, 46 (25%) reported that women in medicine experienced sexual harassment, with trainees and residents most affected [20, 21, 37, 39, 52–54, 74, 75, 78–82, 85, 88, 94, 95, 107–109, 116–119, 121, 126, 127, 130–132, 148, 150, 152, 154, 158, 160, 161, 173, 174, 182, 184, 186, 187, 196]. On the other hand, both male (70%) and female (69%) residents in obstetrics and gynecology (OBGYN) experienced sexual harassment in the USA [66]. Another study [116] in a US medical college reported that one-third of respondents experienced sexual harassment, a finding that was inversely proportionate to the academic rank held: medical students (51.7%), residents/fellows (31%) and faculty members (25%). Similarly, sexual harassment was higher among women in vascular surgery who did not hold leadership or academic titles and were ranked lower than assistant professors [152]. This phenomenon holds true for multiple specialty areas in medicine: respondents from a study in Australia and New Zealand reported a higher proportion of sexual harassment, bullying, and discrimination among female trainees in ophthalmology compared to staff ophthalmologists [132]. An online survey of cardiothoracic surgeons also found higher rates of sexual harassment among trainees [85]. A recent study [184] reports for participants from 28 countries on Bullying, Undermining, and Harassment (BUH) in peripheral vascular disease department, where women's experience of BUH was higher than men (53% vs. 38%). Medical students reported the highest prevalence of BUH (57%) followed by residents (65.7%), fellows (41%), and consultants (37%). Another study in France included midwifery, nursing and medical student reported higher prevalence of GBV for female student (93.7vs. 5.4%). In Canada, a higher proportion of women than men in family medicine experienced intimidation, harassment, or discrimination based on gender, and hierarchy was also identified as a factor [95]. Similarly, in orthopedics, women experienced gender-based harassment and sexual harassment significantly higher than men. Similarly, in orthopedics, women experienced gender-based harassment (98% vs.68%) and sexual harassment (83% vs. 71%) significantly higher than men. In this specific study men represented 72% of the sample [19]. One US study with a large, representative sample (n = 6000) from a national survey reported that greater women's representation within a specialty is associated with lower sexual harassment for both men and women from coworkers and patients [206].

**Clinical routines.**   Workplace routines for all health workers were found to be risk factors for exposure to violence, including longer working hours [5, 151, 156, 161, 171, 226], shifting duties, particularly night shifts [14, 30, 55, 56, 69, 76, 124, 139, 141, 155, 156, 167, 172, 175, 188, 219, 243], and direct patient care [153, 159, 172]. Night shifts were found to be a risk factor for WPV among male nurses in Bangladesh [30], Iran [69], Lebanon [219], Turkey [76], Korea [139], and China [155, 167, 243]. In Saudi Arabia, male nurses working with more than ten staff members were found to be at risk of verbal abuse [72]. Additionally, hours of work and type of position were found to be risk factors for WPV in several studies for women. Working full-time [153], shifting duties [124], overtime/more hours of work [226], and direct

patient care [153, 159, 172] were all associated with higher rates of exposure to violence among female nurses. In the case of medical personnel, male GPs in Australia who worked full-time experienced higher levels of verbal abuse than part-time GPs [12].

**Clinical setting.** Several studies examined specific clinical settings and the risk for physical and non-physical violent incidences. Most incidents occurred in the emergency department (ED) and psychiatric settings [72, 76, 89, 105, 110, 111, 138, 153, 166, 169–172, 178, 181, 188, 202, 203, 221, 232, 243]. Several studies reported female nurses suffering non-physical violence in EDs in China [89, 208] and Ethiopia [111] and in psychiatric units in the USA [153]; physical violence was experienced in pediatric clinics in Turkey [138] and China [208], psychiatric units in USA [203], and in the primary/secondary care facilities in Brazil [151]. Female nurses also experienced both physical and non-physical violence in ED [232] and ICU [214] in Greece, in adult health in Scotland [57] and in psychiatric units in Japan [110] ED in Iran [188], and South Africa [178]. Another study reported female nurses experiencing bullying in the operating rooms and maternity wards [43], and male nurses were reported to experience bullying in medical/surgical units, outpatient clinics, and critical care units [101]. Female physicians also experienced bullying and discrimination in laboratory-based specialties and surgical and medical settings [93]. Male nurses in Jordan [169] and Saudi Arabia [72] experienced verbal violence in EDs, as did male physicians in Turkey [76]. These studies demonstrate that WPV is persistently more prevalent among women and nurses across clinical settings.

Furthermore, several studies reported that more female medical personnel report sexual or gender harassment in male-dominated surgical specialties than in other settings [21, 78, 85, 130, 182]. In the surgical specialties, the prevalence was higher for women compared to men in cardiothoracic surgery [85, 130], pediatric surgery (80%) and neurosurgery [130], and vascular surgery [21]. Similarly, sexual harassment was also experienced by female nurses in public hospitals in China [155], Rwanda [7], Ghana [80], and Japan [110]. In addition, both male and female nursing students in Taiwan [86] and Catalonia [179] experienced sexual violence during university education.

**Ethnicity/nationality.** The nationality or ethnicity of the healthcare professionals also was a factor in the experiences of WPV among nurses and physicians. For instance, male nursing personnel in Iran with non-Farsi ethnicity experienced significantly higher levels of physical violence (OR- 2.34) [115]. Similarly, physical violence was significantly associated with non-Omani and non-Saudi nationality in Oman [71] and Saudi Arabia [171], respectively. In Saudi Arabia, workplace bullying was also more prevalent among expatriate non-Saudi health practitioners [35, 73]. International Medical Graduates (IMG) in Australia, particularly general practitioners and registrars, experienced significantly higher aggression from patients compared to non-IMGs (63% vs. 52%), from relatives (15% vs. 12%) and coworkers (5.7% vs. 3.9%), and this was highest among female IMG staff [114]. In a large academic medical centre in the USA, white female physicians experienced fewer mistreatment episodes than black physicians and those of other races [37]. Further, in radiology, women graduates from foreign medical schools were more likely to report sexual harassment compared to the US graduates (77.1% vs. 54.1%) [82]. Similarly, in China, non-Asian individuals were more likely to experience harassment, and women reported being offered career advancement in exchange for sexual acts [88]. Additionally, bullying was more common among Asians (female faculty members) in a faculty of health sciences in South Africa [58]. Bullying and harassment among non-white vascular physicians were reported in 28 countries [184].

Discrimination based on gender experienced by women, where nationality was a factor, was frequently reported by surgical residents in Australia and New Zealand [94] and the USA [107], and in pediatrics (USA) [121]. While in Canada, there was no significant difference in the proportion of Canadians (46%) and IGMs (41%) in family medicine experiencing

intimidation, harassment, and discrimination (IHD). However, more IMGs perceived IHD based on ethnicity, culture, or language [95]. Similarly, surgical residents (8.8%) in Spain experienced discrimination due to their country of origin, including both women and men [128]. The data revealed an association between WPV and the minority status globally, except for one instance in a public hospital ED in Saudi Arabia, where more Saudis (51.8%) than non-Saudis (33.8%) experienced incidences of all forms of violence [112].

## Discussion

In our comprehensive review of descriptive studies, an apparent gender disparity in the prevalence of workplace violence (WPV) emerged. Overall, 64% of descriptive studies reported a higher prevalence of all forms of WPV for women, including sexual violence, verbal abuse, discrimination, bullying and physical violence. On the other hand, only 17% of the descriptive studies reported men's higher experience in all forms of WPV, including physical violence, verbal violence, bullying and sexual violence. The remaining 19% of the studies that reported higher prevalence for various forms of WPV, either for men or women, are presented in Table 1. All these studies also reported several factors explaining the disparities in prevalence rates for different forms of violence among diverse groups. Firstly, some studies in our review reported insufficient data due to underreporting because of the retrospective nature of reporting mechanisms [83, 156, 166, 203, 232], as most of the incidents were reported after they had occurred, thus introducing the potential for recall bias. Retrospective reporting can also affect the participant's ability to accurately recall the incident because, over time, they may tend to express feelings to friends and family members, which helps alleviate distress. Additionally, the reporting hierarchy in the organization and the research process [83, 136, 232] bring challenges to accurate reporting because of the fear of retaliation by the supervisors, as many were perpetrators of violence [74, 233, 237].

While these factors contributed to variability in data, they also provided insight for addressing gender-based workplace violence and achieving justice for affected individuals, particularly women, which involves multifaceted dimensions. First, it necessitates shielding individuals from existing and potential aggressors by bolstering policies and reporting efforts to safeguard rights in the workplace, such as fostering a comprehensive understanding of safety within the work environment. Secondly, addressing victims' grievances requires strengthening institutional responses tailored to GB-WPV. Lastly, imposing stringent expectations and repercussions on perpetrators entails heightening the consequences for individuals perpetrating such acts and increasing awareness. This emphasizes three critical approaches: enhancing policies, fortifying institutional capacities, and implementing tailored intervention programs for those involved with GB-WPV [244]. In addition, research efforts should focus on understanding barriers to reporting and devising strategies to enhance reporting accuracy, working in tandem with healthcare institutions and supervisors to develop more effective reporting systems and policies that prioritize the well-being and safety of all staff in a way which protects victims of vertical violence. Further, institutions may consider using non-institutional groups to collect and manage information about GB-WPV. Using non-institutional mechanisms may reduce interference by institutional self-interests and reduce gender biases within healthcare [245].

Further, studies in this review reported methodological constraints, including sampling frames (small size, convenience sampling, self-selection, non-representative sample, etc.) [13, 81, 82, 90, 106, 163, 230, 234], which may lead to findings biased toward one group or the other. Furthermore, different assessment methods and measurement tools [87, 92, 122, 125, 166, 223, 228, 235] have been acknowledged to limit the generalizability of results. Additionally, differential operational definitions of terms [82, 95, 101, 104] and their understanding can

limit the reporting and lead to insufficient data. Though we did not critically appraise these studies, or include 'grey literature' sources, we acknowledge that these limitations also limited us to producing a cumulative prevalence in this review. Considering the limitations, we presented the proportions of studies that reported a higher prevalence of WPV for men and women and synthesized factors affecting the disproportionate perveances.

As this is a global study, some regions with limited research capacity are at risk of being omitted from this study. In such contexts, formal studies meeting scientific journal standards may not be feasible, leaving significant gaps in our understanding of GB-WPV prevalence and its impact. One of the studies included in this review sent a worldwide invitation for participation in the study about violence in the health system. Though the study received responses from 110 countries, the researchers excluded responses from 31 countries because of inappropriate responses that did not meet the rigor of the research process [175]. In those contexts, incidents of GB-WPV may be documented in various sources beyond traditional scientific literature, such as internal hospital documents and social media, if documented at all. The reliance on "grey data" introduces its own set of challenges, including issues of reliability, consistency, and accessibility, which this review did not undertake.

A recent systematic review of 253 studies could not determine any significant differences in the prevalence of any form of WPV according to sex, which was attributed to the sample of studies; only 27% of studies included in that review presented the sex-segregated findings [4]. In our scoping review, we report findings from 226 studies that provided sex-segregated data; WPV is a multifaceted topic where women's experience of violence was disproportionately high for almost all forms and contexts. Developing gender-sensitive programs, processes, and policies in healthcare settings is crucial, including a gender-balanced workforce that could benefit both men and women [206]. This approach not only aims to safeguard those from prevalent forms of violence but also acknowledges and addresses the often-understated experiences of violence encountered by men. Training and education sessions have been deemed effective when there is a multidisciplinary approach; they focus on education to enhance knowledge and alter attitudes [244]. These tailored initiatives could help mitigate instances of violence [245]. In addition, the identified gender-based workplace violence (GB-WPV) trends among healthcare professionals should be investigated using rigorous scientific standards to better explore the phenomenon of GB-WPV and related factors [136]. Furthermore, studies must investigate GB-WPV in various clinical settings on a larger scale, including trialing interventions (policies and reporting mechanisms) and their impact by adopting longitudinal, prospective study designs [246]. The revised policies and interventions must consider gender mainstreaming (integrating gendered perspectives in all phases and including both men and women in developing programs and policies) before being implemented in the clinical settings.

Gendered power relations within organizational and professional hierarchies played a critical role in enabling WPV between and within professional groups. For instance, studies that included medical and nursing personnel [13] found that gender is a significant predictor (OR = 9) for WPV in primary care clinics. Female nurses and physicians were 11 times more likely to experience verbal abuse and nine times more likely to experience any form of violence than males. This study also reported that nurses have double the risk of experiencing verbal abuse compared to physicians [13]. The prevalence of gender-based discrimination was also higher for women within the medical profession in most high income countries, including Australia, the USA, and Canada [74, 75, 81, 95, 104, 119], as well as Saudi Arabia and India [79, 108]. These hierarchical gendered relations between men and women reflect those in society at large, and in most cultures and geographic locations, men hold most positions of authority. Preventative measures must be enacted, including robust policies against retaliation and

comprehensive training for supervisors on appropriate behaviour. Reforms must consider and confront broader societal gender-based roles of men and women, often reinforcing power imbalances. Recognizing and challenging societal norms is essential to creating sustainable safeguards within healthcare settings, ensuring a more equitable distribution of power and opportunities for men and women. This could include support programs for those impacted. Interventions which expand access to social support are helpful when addressing issues of abuse [247]. By integrating gender-sensitive approaches and survivor considerations into these reforms, institutions can strive towards fostering a workplace culture that addresses workplace violence and promotes gender equality and inclusivity.

Historically, men have dominated decision-making, leadership roles, and participation in healthcare organizations as a direct result of patriarchal social structures [248]. Male professional domination could explain the higher prevalence of WPV among women in our review in various contexts. In that, men dominated healthcare organizations, specifically medicine, and they also held more institutional power than nurses [248, 249]. Grant et al. [249] explained that women's voices often face suppression within these arenas, influenced by the attitudes prevailing among those in positions of power and the prevalent culture of blaming victims [249]. Similarly, Salles et al. [248] elucidated that the scarcity of women in leadership positions within academic medicine reflects deeply ingrained biases, which are then reinforced by biases favouring men as inherently better leaders. This likely contributes to the disproportionate underrepresentation of women in healthcare leadership roles and less power. Based on the findings on risk factors, there is also a need to understand the interconnected nature of social categorizations and how they intersect with gender to shape the experiences of the health workforce in different situations.

Moreover, WPV involves individuals, groups, and the organization/community. Thus, there is a critical need for policies and interventions to address WPV to target eliminating gender inequality more broadly and to focus interventions at different levels [250]. At the level of the individual, interventions should create awareness about the forms of violence, existing policies and mechanisms for reporting that empower individuals to advocate for themselves [251] and others affected by the incidents [127, 128, 161]. We also recommend improving the structural factors, including physical conditions of work and equitable allocation of women and men in positions of authority in the workplace.

Intervention at the organizational level must target changing the organizational climate, focusing on developing and disseminating zero-tolerance policies [250] comprising transparent and trustworthy reporting mechanisms (regardless of the perpetrator, including patients or family members, supervisor, etc.). Proposed interventions include alert systems [166] and 'hot-lines' [211]. These mechanisms must be paired with clear and consistent action [132] to handle complaints for investigations that follow through with sanctions and penalties to the offender [85, 250].

Moreover, given that gender intersects with other social determinants, strategies must consider these and how they may intersect. For example, the age and professional experience of the victim, the clinical setting, the patients' complexity, the nature of work, and the location of an individual in the organizational hierarchy [113] need deliberate attention for inclusivity. The findings of studies included in this review also call for transformational interventions. For instance, in most cultures, women carry a disproportionate amount of domestic responsibility compared to men and thus may require support to manage their large home and work responsibilities. Flexibility in scheduling and supportive workplace cultures are key to changing the work culture at healthcare institutions. Awareness of the vulnerabilities and pressures on early career professionals who may experience additional pressures, including a higher risk of violence in the workplace and domestic/family pressures at home, is vital to building a sustainable

health workforce. Besides, collaborative efforts must be made to alter the cultural and patriarchal systems that contribute to women's exposure to GB-WPV by creating awareness and condemning GB-WPV through media and strategic advocacy directed at appropriate political, cultural, and religious leaders [252]. Finally, the programs and policies initiated to respond to GB-WPV should be tested empirically for their effectiveness [217], and interventions that are based on evidence must inform policies and procedures [12, 83].

Sexual harassment is a form of violence that has significantly affected women in the health workforce and is enabled by the professional and organizational hierarchies rooted in organizational cultures that provide impunity to perpetrators [253]. In our review, 25% of studies reported harassment significantly affected women in the medical workforce, particularly the trainee medical residents in most contexts [20, 21, 37, 39, 52–54, 74, 75, 78–82, 85, 88, 94, 95, 107–109, 116–119, 121, 126, 127, 130–132, 148, 150, 152, 154, 158, 160, 161]. Women in nursing also experienced sexual harassment [36, 40, 49, 86, 99, 129]. On the other hand, lateral violence or bullying was a significant issue highlighted in nursing and midwifery professions [36, 212, 229]. Considering the hierarchical levels that exist within professions and between professions, interventions must be directed to bring change at each level, including at individual (creating awareness and offering protection), organization (transparent and anonymous system for voicing change, flattened hierarchy and leadership training), and at the system level to prevent accumulation of power at the top. Open, transparent reporting relationships, diversity in career pathways and women's inclusion at all levels of leadership have also been suggested as ways to address organizational hierarchy that may perpetuate GB-WPV [254]. In addition, mandatory training in programs tailored to recognize, manage, and prevent GB-WPV for all healthcare professionals is imperative [251]. Similarly, policy formulation and implementation for preventing and managing WPV at the national level (e.g., Ministry of Health and professional councils and associations) [218, 252] and creating reforms for independent monitoring, reporting, and sanctioning to end impunity [250] are crucial steps. To address this issue, governments, irrespective of geographical location, ought to bolster the legal system's capacity to handle cases of sexual abuse effectively including revising labour laws, introducing special legislation and enforcing the same [3]. Since the guidelines developed in 2002 by ILO, ICN, WHO, and PSI are useful in addressing workplace violence and guiding governments, we suggest the revision and joint efforts of the global health alliances to revise the "Framework Guidelines for Addressing Workplace Violence in the Health Sector" [3] with regards to strengthening legal systems in all counties. A recent analysis of Canadian court cases of violence against nurses revealed that despite having significant injuries, historically, being a nurse was not always considered an aggravating factor in sentencing under criminal law. Therefore, the authors highlighted the need for ongoing legal efforts to combat the widespread acceptance of workplace violence in healthcare and the enactment or stringent enforcement of laws to safeguard victims' rights [255], thereby providing a more robust framework for protection and recourse against WPV.

## Conclusion

Entrenched hierarchical structures often reflect traditional gender norms, where men predominantly hold leadership positions and women are confined to frontline care roles. Simplifying even patient-initiated GB-WPV as a by-product of physical proximity overlooks the deeper systemic issues. Our research reveals how GB-WPV is symptomatic of broader societal injustices rooted in sexism and discrimination, affecting marginalized groups, including women across the globe. These power imbalances create environments where women's voices are marginalized, their concerns dismissed, and their experiences of violence trivialized. This

marginalization not only limits their agency but also exacerbates their vulnerability to GB-WPV. Failing to acknowledge the gendered origins of WPV places countless women in healthcare at risk of experiencing clear violations to personhood and enduring adverse health outcomes and premature career disruptions. The repercussions of GB-WPV resonate throughout the healthcare system, resulting in substantial provider attrition, compromised patient care, and an overburdened healthcare infrastructure struggling to meet the needs of society. We acknowledge that looking at a single analytical category, such as gender, negates the complex ways in which other social categories influence experiences of WPV. Further evaluation is needed to understand the interconnected nature of social categories such as race, gender, sexual orientation, socioeconomic status, ethnicity, immigration status, and more, as well as how they intersect to shape the experiences of WPV.

## Supporting information

**S1 Appendix. Definitions of various forms of WPV.**
(PDF)

**S2 Appendix. Registered protocol for WPV among health workforce based on victims gender.**
(PDF)

**S3 Appendix. PRISMA-ScR filled-checklist.**
(PDF)

**S4 Appendix. Search strategy for the scoping review.**
(PDF)

## Acknowledgments

The authors would like to acknowledge Ms. Mikaela Gray, a Librarian in health sciences at the University of Toronto, for assisting with the search strategy development; and Adam Fuseini, a PhD student at UofT for assistance with software.

## Author Contributions

**Conceptualization:** Sioban Nelson.

**Data curation:** Basnama Ayaz.

**Formal analysis:** Basnama Ayaz, Andrea L. Baumann, Graham Dozois.

**Investigation:** Sioban Nelson, Basnama Ayaz, Andrea L. Baumann, Graham Dozois.

**Methodology:** Basnama Ayaz, Andrea L. Baumann.

**Project administration:** Basnama Ayaz.

**Resources:** Basnama Ayaz.

**Software:** Graham Dozois.

**Supervision:** Sioban Nelson.

**Validation:** Sioban Nelson, Basnama Ayaz, Andrea L. Baumann, Graham Dozois.

**Writing – original draft:** Basnama Ayaz, Andrea L. Baumann.

**Writing – review & editing:** Sioban Nelson, Graham Dozois.

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
