## [Decision Letter · Decision Letter 0]

21 Feb 2024

PGPH-D-23-02487

A Gender-Based Review of Workplace Violence Amongst the Global Health Workforce—A Scoping Review of the Literature

Dear Dr. Basnama Ayaz

Thank you for submitting your manuscript to PLOS Global Public Health. After careful consideration, we feel that it has merit but does not fully meet PLOS Global Public Health’s publication criteria as it currently stands. Therefore, we invite you to submit a revised version of the manuscript that addresses the points raised during the review process.

We look forward to receiving your revised manuscript.

Kind regards,

Nazik Hammad, MD, FACP, FRCPC

Academic Editor

Journal Requirements:

2. We have noticed that you have uploaded Supporting Information files, but you have not included a list of legends. Please add a full list of legends for your Supporting Information files after the references list.

3. In the online submission form, you indicated that "The manuscript is a scoping review of workplace violence among healthcare professionals based on gender. All the abstracted data pertinent to the manuscript (190 studies) is tabulated in Table 1 and Table 2. References are also included for all the sources used in the manuscript. The search strategy is included in Appendix B for one database, which was translated into all other databases. Data for all screen studies is available from the corresponding author in a citation manager, Zotero". All PLOS journals now require all data underlying the findings described in their manuscript to be freely available to other researchers, either 1. In a public repository, 2. Within the manuscript itself, or 3. Uploaded as supplementary information.

Additional Editor Comments (if provided):

Please address the suggestions made by some of the reviewers

Reviewers' comments:

Reviewer's Responses to Questions

**Comments to the Author**

1. Does this manuscript meet PLOS Global Public Health’s publication criteria? Is the manuscript technically sound, and do the data support the conclusions? The manuscript must describe methodologically and ethically rigorous research with conclusions that are appropriately drawn based on the data presented.

Reviewer #1: Yes

Reviewer #2: Yes

Reviewer #3: Yes

Reviewer #4: Yes

2. Has the statistical analysis been performed appropriately and rigorously?

Reviewer #1: N/A

Reviewer #2: N/A

Reviewer #3: N/A

Reviewer #4: Yes

3. Have the authors made all data underlying the findings in their manuscript fully available (please refer to the Data Availability Statement at the start of the manuscript PDF file)?

Reviewer #1: Yes

Reviewer #2: Yes

Reviewer #3: Yes

Reviewer #4: Yes

4. Is the manuscript presented in an intelligible fashion and written in standard English?

Reviewer #1: Yes

Reviewer #2: Yes

Reviewer #3: Yes

Reviewer #4: Yes

5. Review Comments to the Author

Reviewer #1: The authors address a very important and neglected topic of healthcare and healthcare workforce research. The methods are well justified, transparent and well described; the results are comprehensive and well presented. Major benefits of this scoping review are its broad inclusive definition of gender-based workplace violence and the analysis of a large number of studies form different geographical regions across the globe and different healthcare systems and organisational settings that provides a more complex and nuanced picture of gender-based violence at the workplace. Another important benefit is an intersectional approach that highlights that women healthcare workers are overall threatened the most by violence at the workplace, but other lines of social inequality, like race/ethnicity, intersect and may exacerbate the threats. The authors are also sensitive to the limitations of binary gender approach that dominated the studies included in the review.

The paper contributes much needed new knowledge to a very important topic. I read the manuscript carefully and with great interest but do not see any issues that would need further improvement. I think the manuscript should be published as soon as possible.

Reviewer #2: The scoping review aims to report on the prevalence and risk factors of gender-based workplace violence in healthcare settings. This review provides a comprehensive picture about a very important issue but always being neglected. The searching strategies were described in detail. A total of 190 articles were included for analysis. The summarized results are presented clearly with a logical flow. The authors have done an excellent job on providing recommended strategies at individual, organizational and system levels for addressing gender-based workplace violence. Well done! I would recommend this scoping review for publication after addressing the following minor comments:

Page 4 line 87-88:- please report the prevalence of both nurses and physicians in Barbados (Ref13)

Line 98-101:- gender-based violence consists of both men and women. However, the definition here is for women only which may cause discrimination for men. Suggest using another definition which uses ‘individuals / persons’ rather than ‘women’

P.5 Line 106-109, Influence of gender on workplace violence

Apart from the references included in this section, the recent Lancet commission report – women, power and cancer also highlight the impact of gendered power relations that disproportion impact women. Suggest adding this most recent report as a reference to strengthen the argument here.

Reference:

Ophira Ginsburg, Verna Vanderpuye, Ann Marie Beddoe, Nirmala Bhoo-Pathy, Freddie Bray, Carlo Caduff, Narjust Florez, Ibtihal Fadhil, Nazik Hammad, Shirin Heidari, Ishu Kataria, Somesh Kumar, Erica Liebermann, Jennifer Moodley, Miriam Mutebi, Deborah Mukherji, Rachel Nugent, Winnie K W So, Enrique Soto-Perez-de-Celis, Karla Unger-Saldaña, Gavin Allman, Jenna Bhimani, María T Bourlon, Michelle A B Eala, Peter S Hovmand, Yek-Ching Kong, Sonia Menon, Carolyn D Taylor, Isabelle Soerjomataram, Women, power, and cancer: a Lancet Commission, The Lancet, 2023, https://doi.org/10.1016/S0140-6736(23)01701-4. ISSN 0140-6736.

P.9 line 218: change ‘countries’ to ‘countries/special regions’ as Taiwan is not a country. Change ‘Palestine’ to ‘the State of Palestine’ which is the official name more commonly used.

Reviewer #3: This article is an important contribution to gathering global evidence on gender-based workplace violence. It includes a broad range of sources and presents a comprehensive overview of reports of WPV in different settings. It is timely that the gender aspects of WPV are being given greater attention in such a thorough way. The article will be useful to researchers, health care providers, civil society organisations and advocates.

The authors may want to consider:

- Reframing the recommendation at lines 618-621, relating to strengthening legal systems' response to sexual abuse, should be broader. Suggest reviewing the recommendations on reform/ strengthening of legal systems of the "Framework Guidelines for Addressing Workplace Violence in the Health Sector", eg in section 2.1 relating to Government. There are a range of recommendations in this framework which remain highly relevant to understanding and addressing GB-WPV.

- Reviewing the conclusion to ensure it has not overly simplified the results of the report. For example, consider rephrasing the sentence, "Women comprise most of the health workforce, are clustered in frontline care

roles and tend to be lower in the professional and organizational hierarchy. As a result of these factors, female healthcare providers are exposed to a higher prevalence of GB- WPV." This seems to undercut the idea that women face greater discrimination and violence fundamentally due to their gender - earlier in the article, gendered power relations and instrinsically patriarchal structures are more clearly acknowledged.

- Consider rephrasing the sentence, "Without attention to the gendered basis of WPV, millions of women in healthcare are at increased risk of poor health outcomes and their careers are potentially cut short due to WPV." The authors may want to reference the personal and psychological harms and breaches of human rights which women in healthcare may experience due to WPV.

- Consider making recommendations for further research / data required to better understand other intersectionalities which are relevant for WPV - eg sexuality, race, etc, noting that these intersectionailities are acknowledged earlier in the report.

Reviewer #4: I congratulate the authors on their work. This was an immense effort given the limitations in the literature available on this topic, but the authors were able to draw sound conclusions and provide concrete interventions on multiple levels to address GB-WPV. My only comment is that there are many parts of the world where GB-WPV occurs but research capacity is very limited, and therefore having a formal study on it that would be published in a scientific journal (and therefore be in the databases included in this study), would be impossible. Incidents of GB-WPV, including the circumstances surrounding them can be reported in internal documents of hospitals/health systems, on social media, and in the print and non-print news, and not necessarily in scientific journals. Unfortunately, there are even instances when GB-WPV is not reported at all for multiple reasons. This is an inherent limitation of undertaking a study on this topic. Therefore, the authors could perhaps include a statement that it is likely that the prevalence of GB-WPV and the gravity of the situation, is under-reported and understated, and that we may only be scratching the surface of the ground truth. This could be tied in to their recommendations that further and more rigorous studies are needed on GB-WPV.

6. PLOS authors have the option to publish the peer review history of their article (what does this mean?). If published, this will include your full peer review and any attached files.

**Do you want your identity to be public for this peer review?** For information about this choice, including consent withdrawal, please see our Privacy Policy.

Reviewer #1: No

Reviewer #2: No

Reviewer #3: No

Reviewer #4: No

While revising your submission, please upload your figure files to the Preflight Analysis and Conversion Engine (PACE) digital diagnostic tool, https://pacev2.apexcovantage.com/. PACE helps ensure that figures meet PLOS requirements. To use PACE, you must first register as a user. Registration is free. Then, login and navigate to the UPLOAD tab, where you will find detailed instructions on how to use the tool. If you encounter any issues or have any questions when using PACE, please email PLOS at figures@plos.org. Pl

---

## [Editor Report · Decision Letter 1]

21 May 2024

A Gender-Based Review of Workplace Violence Amongst the Global Health Workforce—A Scoping Review of the Literature

PGPH-D-23-02487R1

Dear Basnama Ayaz, BScN, MScN, PhD (

We are pleased to inform you that your manuscript 'A Gender-Based Review of Workplace Violence Amongst the Global Health Workforce—A Scoping Review of the Literature' has been provisionally accepted for publication in PLOS Global Public Health.

Best regards,

Nazik Hammad, MD, FACP, FRCPC

Academic Editor